# Behavioral discrimination and olfactory bulb encoding of odor plume intermittency

Ankita Gumaste[1,2,3], Keeley L Baker[2,3], Michelle Izydorczak[2], Aaron C True[4], Ganesh Vasan[2], John P Crimaldi[4], Justus Verhagen[1,2,3]*

[1]Interdepartmental Neuroscience Program, Yale University, New Haven, United States; [2]John B. Pierce Laboratory, New Haven, United States; [3]Department of Neuroscience, Yale School of Medicine, New Haven, United States; [4]Department of Civil, Environmental and Architectural Engineering, University of Colorado, Boulder, United States

**Abstract** In order to survive, animals often need to navigate a complex odor landscape where odors can exist in airborne plumes. Several odor plume properties change with distance from the odor source, providing potential navigational cues to searching animals. Here, we focus on odor intermittency, a temporal odor plume property that measures the fraction of time odor is above a threshold at a given point within the plume and decreases with increasing distance from the odor source. We sought to determine if mice can use changes in intermittency to locate an odor source. To do so, we trained mice on an intermittency discrimination task. We establish that mice can discriminate odor plume samples of low and high intermittency and that the neural responses in the olfactory bulb can account for task performance and support intermittency encoding. Modulation of sniffing, a behavioral parameter that is highly dynamic during odor-guided navigation, affects both behavioral outcome on the intermittency discrimination task and neural representation of intermittency. Together, this work demonstrates that intermittency is an odor plume property that can inform olfactory search and more broadly supports the notion that mammalian odor-based navigation can be guided by temporal odor plume properties.

*For correspondence: jverhagen@jbpierce.org

Competing interest: The authors declare that no competing interests exist.

## Editor's evaluation

This important work addresses the novel question for the vertebrate olfactory community of whether mice can discriminate odorant intermittency. The evidence supporting the conclusions is convincing. The authors used multiple experimental and analytical tools. The work will be of interest to sensory physiologists, both working in olfaction and navigation.

## Introduction

When navigating an airborne odor landscape, animals interact with highly dynamic and diverse odor structures (*Fackrell and Robins, 1982*; *Crimaldi and Koseff, 2001*; *Crimaldi et al., 2002*; *Connor et al., 2018*). Variations of odor plume parameters can, for example, create plumes with diffusive odor signals and little fluctuation, as well as plumes where odor is pulled into filaments ('odor whiffs') that are interleaved with layers of odor-free air. Despite these complex odor environments, many animals are adept at locating odor sources within airborne plumes (*Vickers, 2000*; *Bhattacharyya and Bhalla, 2015*; *Gire et al., 2016*; *Baker et al., 2018*; *Gumaste et al., 2020*). Odor plume characteristics can be measured using plume statistics such as concentration distribution, odor whiff frequency, and odor

intermittency, here defined as the fraction of time odor is present at a sampled point within odor plume space (*Yee et al., 1993*; *Justus et al., 2002*; *Connor et al., 2018*). Statistical properties of these structures may provide animals with critical information to aid odor source localization (*Boie et al., 2018*; *Reddy et al., 2022*). Although we have a growing understanding of the quantification of features of these odor plumes, the plume properties that mammals use to navigate to an odor source remain largely unknown. Several temporal properties of odor plumes change with distance from the odor source and therefore serve as candidate properties that may be used for odor-guided navigation (*Balkovsky and Shraiman, 2002*; *Vergassola et al., 2007*; *Schmuker et al., 2016*; *Michaelis et al., 2020*).

Changes in odor plume temporal properties affect both insect and rodent navigation strategies (*Reddy et al., 2022*). One such temporal property is odor intermittency, which we focus on in the present study. Moths fly faster and straighter upwind within odor plumes with lower odor plume intermittency (*Mafra-Neto and Cardé, 1994*; *Vickers and Baker, 1994*). In addition, the frequency and duration of time between odor whiffs, both of which can influence intermittency, affect pausing and orienting behavior of *Drosophila* (*Álvarez-Salvado et al., 2018*; *Demir et al., 2020*). Further probing into this behavior through mathematical models suggests that combining odor intermittency sensing along with detection of other temporal odor plume properties enhances odor source localization in flies (*Jayaram et al., 2022*). While the effect of intermittency on rodent odor-based navigation remains yet to be studied, recent studies have highlighted that rodent navigation strategies also depend on temporal properties of odor plumes, such as variance of odor whiffs and number of odor whiff encounters (*Bhattacharyya and Bhalla, 2015*; *Gumaste et al., 2020*; *Tariq et al., 2021*). Together this indicates the contribution of temporal odor plume features to navigation within an airborne odor plume.

Acquisition of odor information by mammals navigating within complex odor environments can be controlled through active sampling in the form of sniffing (*Wesson et al., 2009*). Rodent sniffing behavior is highly dynamic during laboratory odor-guided tasks, suggesting that animals actively change their sampling strategies to adjust the odor information they process (*Uchida and Mainen, 2003*; *Kepecs et al., 2007*; *Verhagen et al., 2007*; *Wesson et al., 2008*; *Wesson et al., 2009*; *Reisert et al., 2020*). Additionally, during odor plume navigation, sniff frequency is modulated with changes in olfactory search phases, such as initial investigation and odor-approaching phases (*Khan et al., 2012*; *Bhattacharyya and Bhalla, 2015*; *Findley et al., 2021*; *Reddy et al., 2022*). This indicates that even in more naturalistic odor environments, rodent active sampling is highly modulated. Inhalation patterns also affect neural representation of odor stimuli and early olfactory processing within the olfactory bulb. Sniffing influences how odors travel in the epithelium and reach odorant receptors on olfactory sensory neurons (OSNs), influencing resulting neural responses (*Mainland and Sobel, 2006*; *Scott, 2006*). Sustained high-frequency sniffing diversifies and attenuates olfactory bulb neural responses (*Verhagen et al., 2007*; *Díaz-Quesada et al., 2018*; *Jordan et al., 2018a*; *Eiting and Wachowiak, 2020*), as well as elicits changes in olfactory bulb output cells, mitral and tufted cells (M/T cells), firing rate, and response latencies (*Carey and Wachowiak, 2011*; *Jordan et al., 2018b*; *Shusterman et al., 2018*). Thus, sniffing has diverse, but quantifiable effects on early olfactory processing. The intersection between odor environment properties and the modulation of active sampling patterns may inform neural representations of temporal odor plume properties while animals are navigating.

The rodent olfactory system has access to temporal information that may aide in localizing odor sources. Fluctuating odor input can be reliably represented by olfactory bulb glomeruli. In both rats and mice, M/T cells' responses correlate with highly dynamic odor input and the correlation strength depends on the stimulus odor plume statistics (*Gupta et al., 2015*; *Lewis et al., 2021*). This dependence of glomerular response properties on odor plume statistics provides further reason to explore which temporal odor plume properties can be perceptually discriminated and therefore may inform navigation. Behaviorally, mice can discriminate changes in temporal properties of odors such as odor duration and odor whiff frequency (*Li et al., 2014*; *Rebello et al., 2014*; *Ackels et al., 2021*). Additionally, glomerular responses of OSNs and M/Ts can support this temporal discrimination (*Ackels et al., 2021*). Based on olfactory bulb response properties, behavioral discrimination abilities, and dependence of behavioral strategies on odor plume statistics, we hypothesize that mice can use temporal odor plume properties for odor-guided navigation.

Here, we focus on the temporal property of odor intermittency and sought to investigate if mice can use odor intermittency for odor plume navigation. We use a combination of behavioral training and simultaneous calcium imaging of the dorsal olfactory bulb to determine if mice can detect differences in odor intermittency and if the olfactory bulb encodes information that enables intermittency discrimination. Additionally, using an artificial sniffing system we further address how active sampling strategies affect the olfactory bulb representation of fluctuating odor plumes. We found that mice can behaviorally discriminate between odor plume samples of high and low intermittency and that active sampling behavior can predict discrimination success. Additionally, we found that both input and output neurons in the olfactory bulb encode information that allows for the detection of differences in odor intermittency and that intermittency encoding in M/T cells is affected by sniff frequency. We observed heterogeneity in glomerular response properties based on the intermittency of the odor stimulus, which may inform intermittency discrimination and indicate specific glomeruli that best contribute to this discrimination.

## Results

### Behavioral discrimination of intermittency

Several odor plume statistical properties change with increasing distance from the odor source and therefore serve as candidate properties that can be used for odor-based navigation. One such property is intermittency, which measures the fraction of time odor concentration is above a threshold at a sampled point in space (*Crimaldi et al., 2002*; *Connor et al., 2018*). To characterize mouse behavioral and neural responses to fluctuating odor stimuli of varying intermittency, we designed a counterbalanced olfactometer in which the airflow remains constant while odor concentration changes (*Figure 1A–C*, *Figure 1—figure supplement 1A*). We were able to produce odor stimuli that closely follow sample measurements (*Figure 1D and E*, *Figure 1—figure supplement 1B*, *Figure 1D* maximum cross-correlation=0.87 ± 0.12, lag = 160 ms) taken from an acetone-based odor plume using planar laser-induced fluorescence (*Connor et al., 2018*).

To determine if mice could discriminate between fluctuating odor stimuli with varying intermittency values, we trained a cohort of OMP-GCaMP6f and THY1-GCaMP6f mice (expressing GCaMP6f in OSNs and M/T cells, respectively) on a Go/No-Go task to discriminate between low (CS-, intermittency≤0.15) and high intermittency stimuli (CS+, 0.2≤intermittency≤0.8) using methyl valerate, a fruit-associated odor. Mice were tested on this intermittency task using three different stimulus types: (1) Naturalistic in which odor samples were taken directly from the odor plume imaged by *Connor et al., 2018*, and normalized so that all plume traces reach the same maximum concentration. (2) Binary naturalistic, which represent a thresholded version of naturalistic stimuli where odor is either at the maximum concentration or off. (3) Square-wave in which odor pulses of fixed duration and duty cycle are presented (*Figure 1F*). Information-theoretic analysis used to study the odor plume cues that may be informative in determining odor location shows that the resolution of odor concentration representation needs only be coarse, while at strategic increments, for successful navigation (*Boie et al., 2018*). Binary naturalistic stimuli were hence included to test the effect of intermediate concentration changes of the naturalistic stimuli on animal performance. Square-wave stimuli were included to test the effect of the random nature (aperiodicity) and frequency of naturalistic odor whiff presentation on animal performance. The intermittency values of the delivered odor stimuli, as determined by the odor concentration measured with a mini photoionization detector (PID), closely matched the expected intermittency values (*Figure 1G*, *Figure 1—figure supplement 1C*). To test the degree to which odor concentration integration may inform decisions on the intermittency discrimination task, mice were tested on interleaved trials using a two gain values, where in trials with a gain of 0.5, the maximum stimulus concentration was halved (*Figure 1F*, *Figure 1—figure supplement 1D*).

In the Go/No-Go task, CS+ and CS- were presented in a random order preceded by a tone cue. When presented with a high intermittency odor stimulus (CS+), head-fixed mice running on a freely rotating wheel were required to lick during a 1.5 s decision period following the 6 s odor period to receive a water reward (hit). When presented with a low intermittency (CS-), mice were required to withhold licking during the decision period to avoid an increased inter-trial interval (miss, *Figure 2A*, *left, right*). Mouse performance on CS+ trials of the intermittency discrimination task increased as the difference between the CS+ and CS- intermittency values increased, showing that mice can

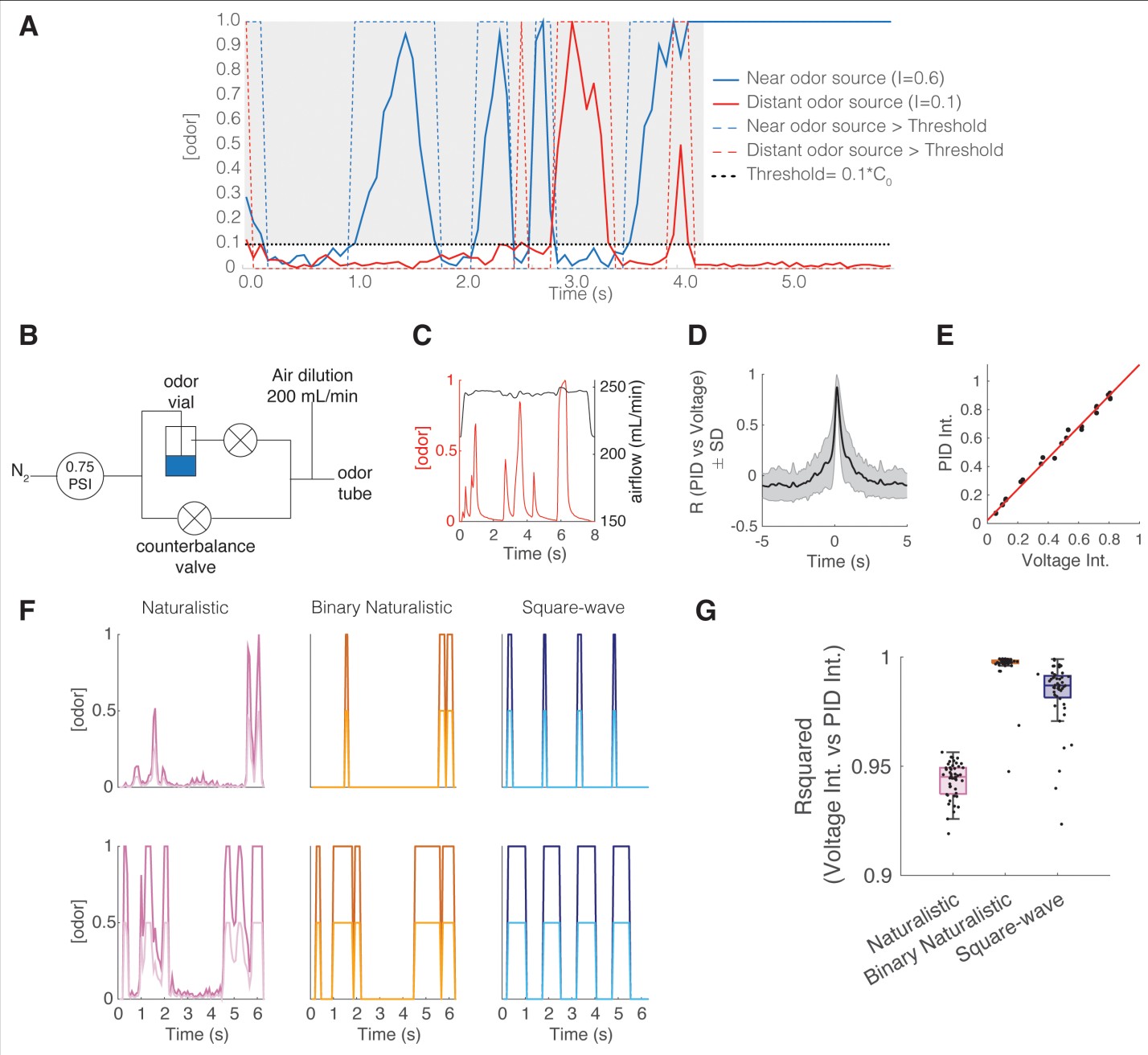

**Figure 1.** Intermittent odor plume stimuli and olfactometer design. (**A**) Graphical illustration of the intermittency measure. Intermittency (I) is the fraction of time an odorant concentration is above a threshold ($0.1*C_0$, where $C_0$ refers to the time-averaged source concentration). In a turbulent plume I drops as a function of distance. Hence, upstream (near the odor source) I tends to be large (here I=0.6) compared to downstream, distant from the odor source (here I=0.09). A steady signal has a high intermittency, and a sporadic signal has a low intermittency. (**B**) Odor delivery system used to deliver methyl valerate and 2-heptanone. Two counterbalanced proportional valves maintained constant flow rate. (**C**) Example of odor concentration (red) and flow rate (black) on a single trial. (**D**) Cross-correlation between photoionization detector (PID) measurement (odor concentration) and the command voltage driving movement of the odor proportional valve. Maximum correlation coefficient is 0.872 ± 0.119 at a lag of 160 ms (n=8643 trials). (**E**) Example correlation between the trial intermittency value measured from the PID reading vs the intermittency value measured from the voltage command for one session (n=64 trials). Linear regression: y=1.09x+0.023, $r^2$=0.996, p<0.0001. (**F**) Example traces of odor concentration at gain 1 (darker colors) and gain 0.5 (lighter colors) for naturalistic, binary naturalistic, and square-wave stimuli. (**G**) Median $r^2$ of the correlation between voltage intermittency and PID intermittency for sessions of naturalistic (red), binary naturalistic (orange), and square-wave (blue) stimuli (n=48 sessions per stimulus type, naturalistic median = 0.945 interquartile range [IQR]=[0.937–0.949], binary naturalistic median = 0.998 IQR=[0.997–0.999], square-wave median = 0.987 IQR=[0.982–0.991]).

The online version of this article includes the following figure supplement(s) for figure 1:

*Figure 1 continued on next page*

**Figure supplement 1.** Additional information on intermittent odor plume stimuli and intermittency calculation.

discriminate intermittency (*Figure 2B*). Animals performed above chance at intermittency values ≥ 0.3 (t-tests with Bonferroni correction, p<0.0001). Animal performance did not differ based on genotype, stimulus type, or the odor used as determined by testing animals using 2-hepatone for the binary naturalistic condition in addition to methyl valerate (*Figure 2—figure supplement 1A*, mixed effects model, n=48 sessions, r²=0.183: performance~intermittency+genotype+stimulus type+gain; main effect of intermittency, p<0.0001; main effect of genotype, p=0.46; main effect of stimulus type, p=0.21; 2. Mixed effects model, binary naturalistic for methyl valerate and heptanone, n=48 sessions per stimulus type, r²=0.179: performance~intermittency+odor; main effect of odor, p=1), showing that intermediate concentration changes and the unpredictable nature of the odor plume did not have an effect on intermittency discrimination.

As mentioned, mice were tested on two gain values to determine the degree to which odor concentration integration affected their task decisions. If mice are solely relying on intensity integration, then halving the odor concentration (gain = 0.5) would be fully equivalent to halving the intermittency at gain = 1, as the amount of total absorbed odorant during a trial would be identical in both cases. *Figure 2B* demonstrates that psychometric curves for gain = 0.5 are not right-shifted versions of gain = 1 by the expected equivalent doubling of intermittency. Indeed, the intermittency discrimination thresholds shifted much less (to 0.4, 0.4, and 0.5 at gain = 0.5) than the expected doubling of 0.3 at gain = 1 (i.e. 0.6 at gain = 0.5). Further, although there was an effect of gain on behavioral performance, animal performance at 0.5 gain was significantly better than a psychometric curve prediction of animal performance solely based on odor concentration integration accordingly (*Figure 2—figure supplement 1B*, mixed effects model, n=48 sessions: performance~intermittency+genotype+stimulus type+gain; main effect of gain, p=0.00013, one-tailed t-test with Bonferroni correction, naturalistic intermittency≥0.3, p<0.0001; binary naturalistic intermittency≥0.5, p<0.0001; square-wave intermittency≤0.8, p<0.0001). Animals' hit rate also significantly decreased when tested on the Go/No-Go task with the odor vial replaced with mineral oil (*Figure 2C*, n=12 mice, two-sample t-test naturalistic: odor hit rate = 0.87 ± 0.01, no odor hit rate = 0.23 ± 0.05, p<0.0001; two-sample t-test binary naturalistic: odor hit rate = 0.89±0.01, no odor hit rate = 0.18±0.07, p<0.0001; two-sample t-test synthetic: odor hit rate = 0.86±0.007, no odor hit rate = 0.23±0.07, p<0.0001), confirming that mice are using odor to perform the task. Additionally, when trials are binned by the number of whiffs per trial, the number of whiffs does not have an effect on trial performance, indicating that mice are not 'counting' whiffs to perform the intermittency discrimination task (*Figure 2—figure supplement 1C*, Spearman correlation, n=48 sessions per stimulus type, p>0.05). Taken together, this suggests that mice, on a behavioral level, are capable of discriminating fluctuating odors based on intermittency.

## Effect of active sampling modulation on intermittency discrimination

Rodent sniffing is highly dynamic during odor exploration and odor-based navigation (*Wesson et al., 2008*; *Khan et al., 2012*). We evaluated active sampling during the discrimination task by measuring sniffing in real time using a pressure sensor inserted into the odor tube (*Figure 2A*, *right*). To assess how trial behavior and performance depend on active sampling over the 6 s dynamic odor stimulus, we calculated an estimate of perceived odor intermittency over time by mice. Here, we directly apply the concept of intermittency to the odor sampled during sniffing, thereby retaining a coherent calculation without introducing new assumptions. We defined estimated perceived odor as the odor concentration, as measured by the PID, only during sniff inhalation periods (*Figure 3A*). Using a thresholded version of the estimated perceived odor (see Methods), we estimated the intermittency perceived by animals based on their active sampling behavior. Average cumulative estimated perceived intermittency varied across the 6 s trial, however estimated perceived intermittency naturally separates by odor intermittency by the end of the trial (*Figure 3B*). To assess the effect of odor intermittency and estimated perceived intermittency on decision-making, we quantified the time at which each mouse first licked during all trials by intermittency value. Anticipatory licking is used as a measurement of motivation and early decision-making (*Berditchevskaia et al., 2016*). Mice lick earlier for high intermittency trials based on both odor intermittency and estimated perceived intermittency (*Figure 3C and D*; binary naturalistic: linear regression first lick time vs odor intermittency,

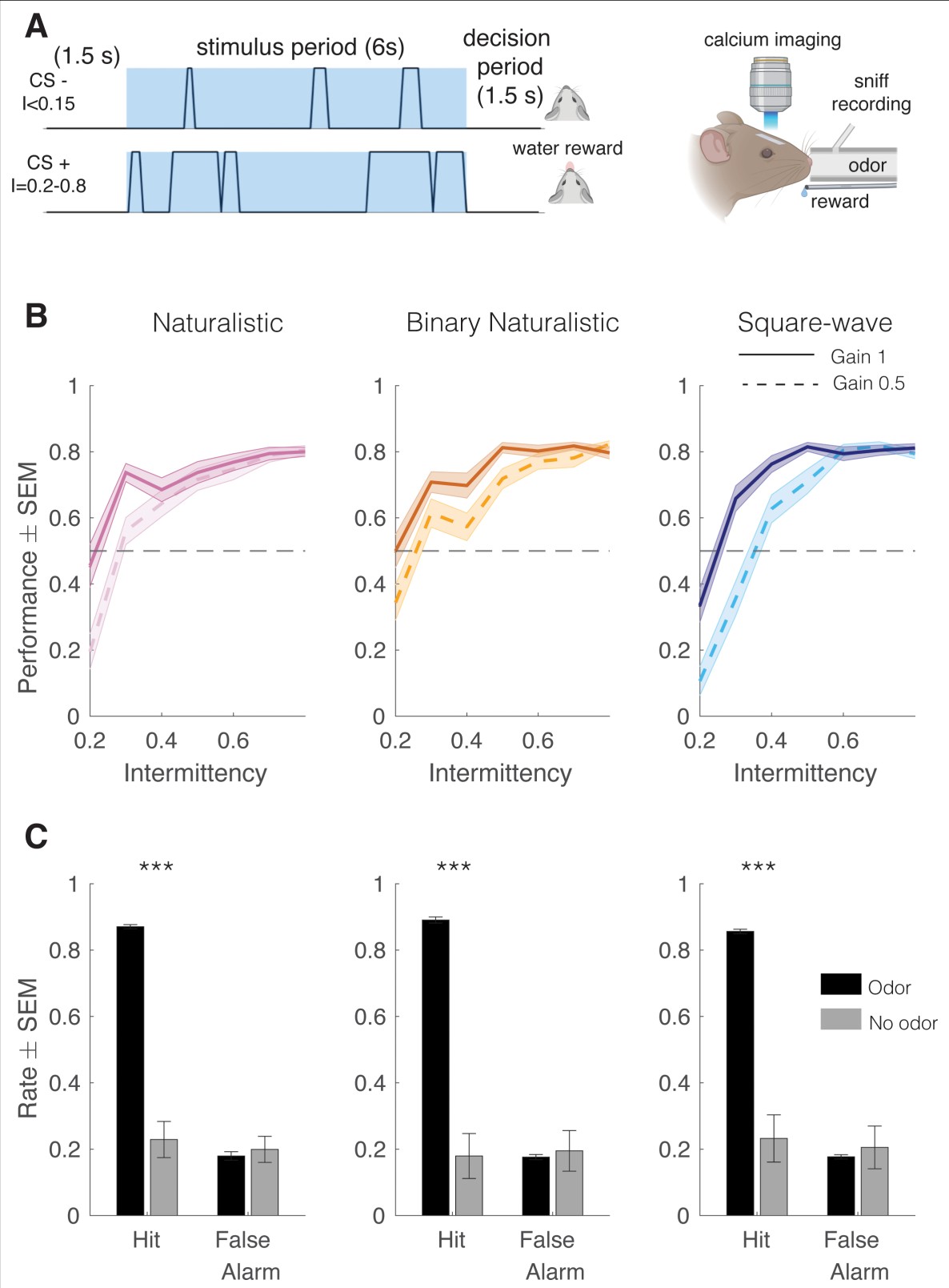

**Figure 2.** Mice can discriminate between fluctuating odor stimuli based on intermittency values. (**A**) Go/No-Go intermittency discrimination task structure. Animals are presented with a 6 s odor stimulus following a 1.5 s delay and after the odor presentation have a 1.5 s decision period during which, if they lick for a CS+, they receive a water reward, and if they lick for a CS-, they receive a punishment in the form of an increased ITI. Left: Imaging and odor delivery setup. Mice are delivered odor through a tube in front of their nose and sniffing is recorded through a pressure sensor

*Figure 2 continued on next page*

*Figure 2 continued*

inserted into the odor tube. Glomerular activity in the dorsal olfactory bulb is imaged using wide-field calcium imaging. (**B**) Mouse performance on intermittency discrimination task. At gain 1, mice perform significantly above chance at intermittency values of 0.3 and above (one-tailed t-test, Bonferroni correction, p<0.0001, n=48 sessions) for all stimulus types. At gain 0.5, mice perform above chance at intermittency values 0.4 and above, naturalistic and square-wave, 0.5 and above, binary naturalistic (one-tailed t-test, Bonferroni correction, p<0.0001, n=48 sessions). (**C**) Hit rates (HR) and false alarm (FA) rates of mice performing the intermittency discrimination task with and without odor. Two-sample t-tests. Naturalistic: $\mu_{HROdor}$=0.87±0.006, $\mu_{HRNoOdor}$=0.23±0.055, p<0.0001, $\mu_{FAOdor}$=0.18±0.013, $\mu_{FANoOdor}$=0.20±0.039, p=0.64, binary naturalistic: $\mu_{HROdor}$=0.89±0.009, $\mu_{HRNoOdor}$=0.18±0.068, p<0.0001, $\mu_{FAOdor}$=0.18±0.008, $\mu_{FANoOdor}$=0.19±0.061, p=0.75, square-wave: $\mu_{HROdor}$=0.86±0.007, $\mu_{HRNoOdor}$=0.23±0.071, p<0.0001, $\mu_{FAOdor}$=0.18±0.006, $\mu_{FANoOdor}$=0.21±0.065, p=0.67.

The online version of this article includes the following figure supplement(s) for figure 2:

**Figure supplement 1.** Intermittency discrimination performance by genotype, odor, and whiff number.

y=−2.25x+5.42, p<0.0001, $r^2$=0.07; binary naturalistic: linear regression first lick time vs estimated perceived intermittency, y=−2.15x+5.53, p<0.0001, $r^2$=0.05; square-wave: linear regression first lick time vs odor intermittency, y=−0.93x+4.58, p<0.001, $r^2$=0.01; square-wave: linear regression first lick time vs estimated perceived intermittency, y=−0.32x+4.4, p<0.001, $r^2$=0.01). For estimated perceived intermittency values that exceed the range of odor intermittency values, the time of first lick continues to decrease. Taken together, this suggests that decision-making or motivation on CS+ trials is dependent on the difference between CS+ and CS- trial intermittency values, as well as confirms the validity of estimated perceived intermittency.

To assess the effect of estimated perceived intermittency on trial outcome, we separated CS+ trials into hit and miss trials and compared the estimated perceived intermittency of these trials across odor intermittency values. There was an interaction between odor intermittency and trial outcome (hit or miss) on the average estimated perceived intermittency (generalized linear model, estimated PI~intermittency*outcome, binary naturalistic: interaction intermittency*outcome, p<0.0001, square-wave: interaction intermittency*outcome, p<0.0001). On trials with intermediate odor intermittency values of 0.4 and 0.5, at which animals initially start performing above chance (*Figure 1H*), animals had a lower estimated perceived intermittency on miss trials when compared to hit trials (*Figure 3E*, two-sample t-tests hit trials estimated perceived intermittency vs miss trials estimated perceived intermittency, binary naturalistic: intermittency 0.4, p=0.02, intermittency 0.5, p=0.04). Additionally, we assessed if animals were simply sniffing at higher average frequencies during odor presentation on hit when compared to miss trials, specifically for intermediate odor intermittencies. Neither trial outcome nor intermittency influenced average trial sniff frequency (*Figure 3—figure supplement 1*). Additionally, animals show a greater increase in pupil dilation and greater decrease in running speed at the onset of the reward period on hit trials when compared to miss trials (*Figure 3—figure supplement 2A–F*). Sniff frequency, pupil dilation, and running speed are more strongly correlated on hit trials when compared to miss trials (*Figure 3—figure supplement 2G–I*), possibly indicating switches in behavioral state (*Findley et al., 2021*).

To test the ability of an animal's estimated perceived intermittency on a given trial to predict trial identity (CS+ or CS-), we trained a linear classifier using trial estimated perceived intermittency values to discriminate between CS+ and CS- trials. As more trial time is added to the estimated perceived intermittency, prediction accuracy increases for all trial intermittency values. Prediction accuracy is significantly above, and is maintained above, the shuffled control earlier in the trial for high intermittency trials when compared to low intermittency trials (*Figure 3F*, two-sample one-tailed t-test with Bonferroni correction, see figure and caption for statistics). This finding provides further support for the notion that as the estimated perceived difference between CS+ and CS- intermittency values increase, animals are more likely to lick earlier in the trial. These findings reiterate the decrease in uncertainty in determining trial identity, as measured by the lick time, at higher intermittency values.

## Spatial mapping of glomerular response properties

We next investigated how neural responses in early olfactory processing can support intermittency discrimination behavior. To do so, we used wide-field imaging to measure glomerular responses in the dorsal olfactory bulb of OMP-GCaMP6f mice, in which GCaMP6f is expressed in OSNs, in awake, behaving animals (*Figure 2A*, *right*). Recently *Lewis et al., 2021*, have shown that glomerular responses track odor plume dynamics with varying strength and that this odor tracking depends

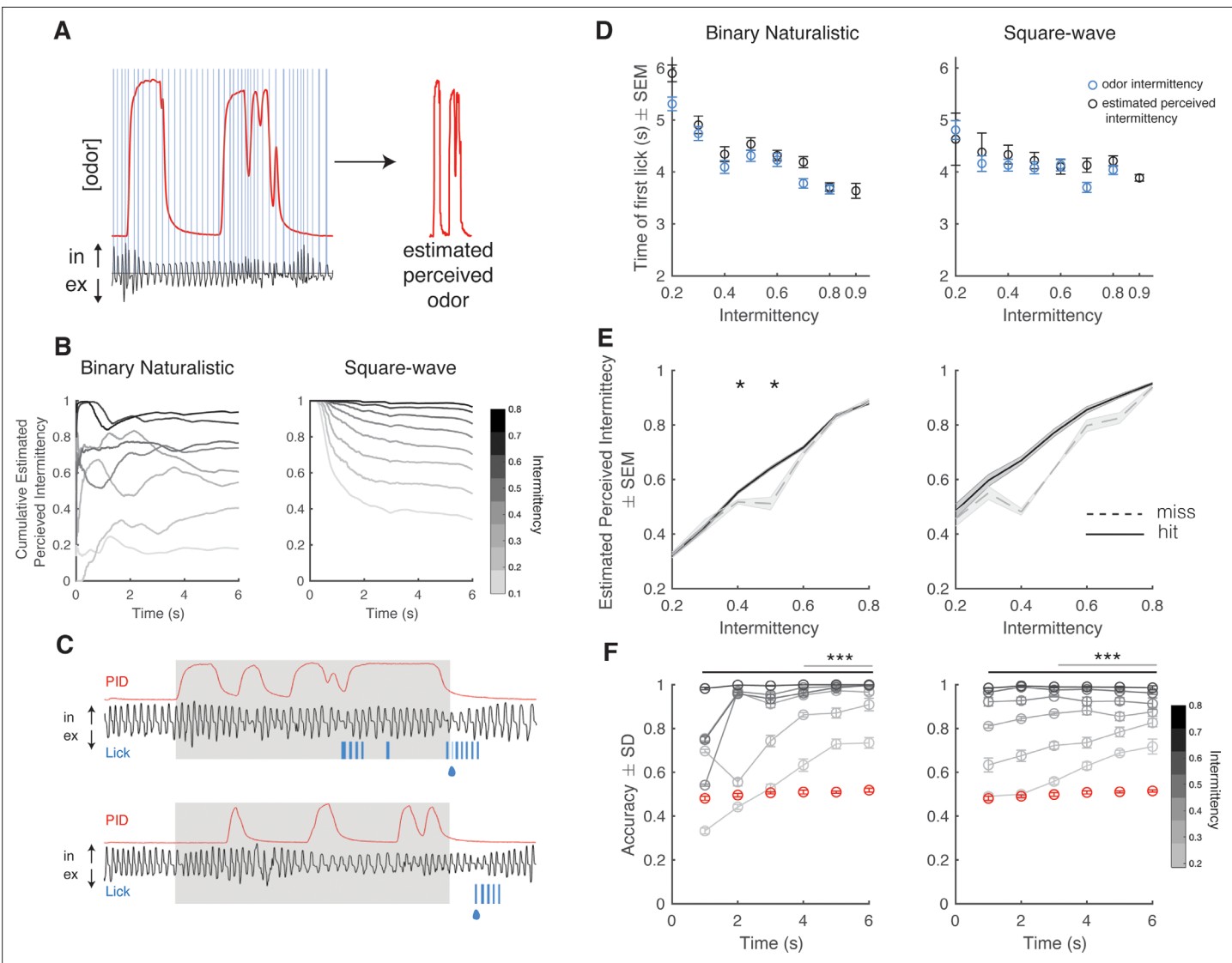

**Figure 3.** Estimated perceived intermittency differs based on trial outcome. (**A**) Example photoionization detector (PID) trace (red) and pressure sensor trace (black), and blue lines correspond to inhalation periods. Left: Example estimated perceived odor trace (PID trace sampled during inhalation periods). (**B**) Average cumulative estimated perceived intermittency (based on estimated perceived odor) across trial time for trials with intermittency values between 0.1 and 0.8 for binary naturalistic (n=1362 trials, *left*) and square-wave (n=1341 trials, *right*). (**C**) Example PID reading (red), sniff trace (black), lick trace (blue) during an example high intermittency trial (*top*, intermittency = 0.8) and low intermittency trial (*bottom*, intermittency = 0.3). Gray area indicates 6 s odor stimulus period. Following the stimulus period is the decision period where a water reward is delivered if animals lick for a CS+ (indicated by the water droplet). (**D**) Time of first lick binned by intermittency using both odor intermittency (blue) and estimated perceived intermittency (black) for binary naturalistic (n=1362 trials, *left*) and square-wave (n=1341 trials, *right*). (**E**) Estimated perceived intermittency vs odor intermittency on hit and miss CS+ trials (n=48 sessions). (**F**) Right: Square-wave. Accuracy of linear classifier performance in predicting trial identity (CS+ or CS-) for trials of intermittency values between 0.2 and 0.8 (CS+) based on estimated perceived intermittency (gray to black lines). Shuffled control is shown in red. One-sided two-tailed t-test with Bonferroni correction. Left: Binary naturalistic, intermittency values ≥ 0.3, all times are significantly above shuffled control (black bar, p<0001). Intermittency values = 0.2, times ≥4 s are significantly above shuffled control (gray line, p<0001). Right: Square-wave, intermittency values ≥ 0.3, all times are significantly above shuffled control (black bar, p<0001). Intermittency values = 0.2, times ≥3 s are significantly above shuffled control (gray line, p<0001) (n=20 repeats per time bin).

The online version of this article includes the following figure supplement(s) for figure 3:

**Figure supplement 1.** Average trial sniff frequency vs odor intermittency on hit and miss trials.

**Figure supplement 2.** Pupil dilation and running speed differ between hit and miss trials.

on odor plume statistics. We wanted to explore if odor intermittency had an effect on glomerular tracking of odor plumes, by measuring the cross-correlation between glomerular responses and PID odor reading of methyl valerate across trials of increasing intermittency (*Figure 4A*). To characterize the spatial mapping of glomeruli based on stimulus tracking, we found the relationship between the odor-response correlation and glomerular location on the surface of the olfactory bulb. When presented with methyl valerate, the glomeruli in the posterior-lateral region of the olfactory bulb had the highest average correlation with the stimulus (*Figure 4B and C*) and this spatial organization is weaker for lower intermittency stimuli in the medial to lateral direction (*Figure 4—figure supplement 1A*, x, intermittency; y, medial to lateral odor correlation, y=0.22x+0.21, $r^2$=0.08, p<0.0001). Additionally, the glomeruli in the posterior-lateral region of the olfactory bulb also have the largest amplitude response and are the fastest to respond to the odor (have the shortest time post sniff onset to reach 75% of the maximum dF/F value corresponding to that sniff, T75, *Figure 4D*, *Figure 4—figure supplement 1B*). If glomeruli are clustered based on their T75, slower responding glomeruli do not track high intermittency stimuli as well as faster responding glomeruli (*Figure 4E*, ANCOVA, CI = [0.245 0.409], Δ in slope: 0.328, p<0.0001). Together, this characterization of glomerular responses confirms the finding of *Lewis et al., 2021*, that glomeruli track fluctuating odor stimuli to different degrees. Additionally, these findings suggest that the glomerular ability to track odor stimuli depends on spatial patterning and that glomerular responses to odors of varying intermittency depend on intrinsic glomerular properties (such as T75).

## Glomerular subpopulations encode differing representations of intermittency

To investigate how OSNs encode stimuli of varying intermittency, we calculated a glomerular intermittency (GI) value for each glomerulus for all trials. Just as we applied the concept of intermittency to the calculation of estimated perceived intermittency based on odor sampling patterns, here we directly applied an intermittency calculation to glomerular responses. GI was calculated by measuring the fraction of time the z-scored glomerular response trace, relative to glomerular-specific non-odor period background activity, was above a z-score threshold of 2 during the odor stimulus period (*Figure 5A*, *right*). We found that glomeruli display diverse intermittency representation across stimuli of varying intermittency based on GI and sought to identify if groups of glomeruli existed based on their GI response properties (*Figure 5A*, *left*). To do so, hierarchical clustering on the inter-glomerular correlation of GI across odor intermittency was performed (*Figure 5B*). To elaborate, in measuring an inter-glomerular correlation, two glomeruli that both show an increase in their GI value across trials of increasing odor intermittency would have a high inter-glomerular correlation. Glomeruli clustered into two groups based on their GI representation of odor intermittency. Glomeruli in clusters 1 and 2 showed inverse and positive relationships, respectively, between GI and odor intermittency. Thus, the average slope between GI and odor intermittency for each glomerulus, GI slope, for cluster 1 was negative and was positive for cluster 2 (*Figure 5Ci*, two-sample Welch's t-test; GI slope, $\mu_{cluster1}$ = –0.59 ± 0.04, $\mu_{cluster2}$ = 0.89 ± 0.02, p<0.0001). Additionally, glomeruli in these two clusters differed in their T75 as well as their average odor correlation (*Figure 5Cii–iv*, *Figure 5—figure supplement 1*, two-sample Welch's t-test; T75, $\mu_{cluster1}$ = 179 ± 5.6 ms, $\mu_{cluster2}$ = 159±0.3 ms, p=0.0024; odor correlation, $\mu_{cluster1}$ = 0.24 ± 0.017, $\mu_{cluster2}$ = 0.31±0.01, p<0.0001).

Biophysical diversity and heterogeneity in neural populations enhances information encoding in the olfactory bulb (*Tripathy et al., 2013*). To test if the heterogeneous population including cluster 1 and 2 of glomeruli can predict intermittency better than a homogeneous population including only one of the two clusters, we used a 20 times threefold cross-validated linear classifier to predict trial identity (CS+ or CS-) using GI values. A homogeneous population including only glomeruli belonging to cluster 1 or cluster 2 predict trial identity with high accuracy (cluster 1: $\mu_{10glomeruli}$ = 0.87±0.01, $\mu_{25glomeruli}$ = 0.87±0.01; cluster 2: $\mu_{10glomeruli}$ = 0.88±0.01, $\mu_{25glomeruli}$ = 0.90±0.01). However, the heterogeneous population, which includes glomeruli of both clusters, shows slightly, but consistently and significantly, higher prediction accuracy using the linear classifier (*Figure 5D*, ANOVA [10–25 glomeruli]: p<0.0001, maximum accuracy for each number of glomeruli added to model: $Max_{10\ glomeruli}$=40% cluster 1, $Max_{15\ glomeruli}$=50% cluster 1, $Max_{20\ glomeruli}$=60% cluster 1, $Max_{25\ glomeruli}$=70% cluster 1). Glomeruli have varying strengths in GI slope, and we sought to understand if the strength of GI slope has an effect on the ability of a glomerulus to contribute to trial identity prediction. Using a linear

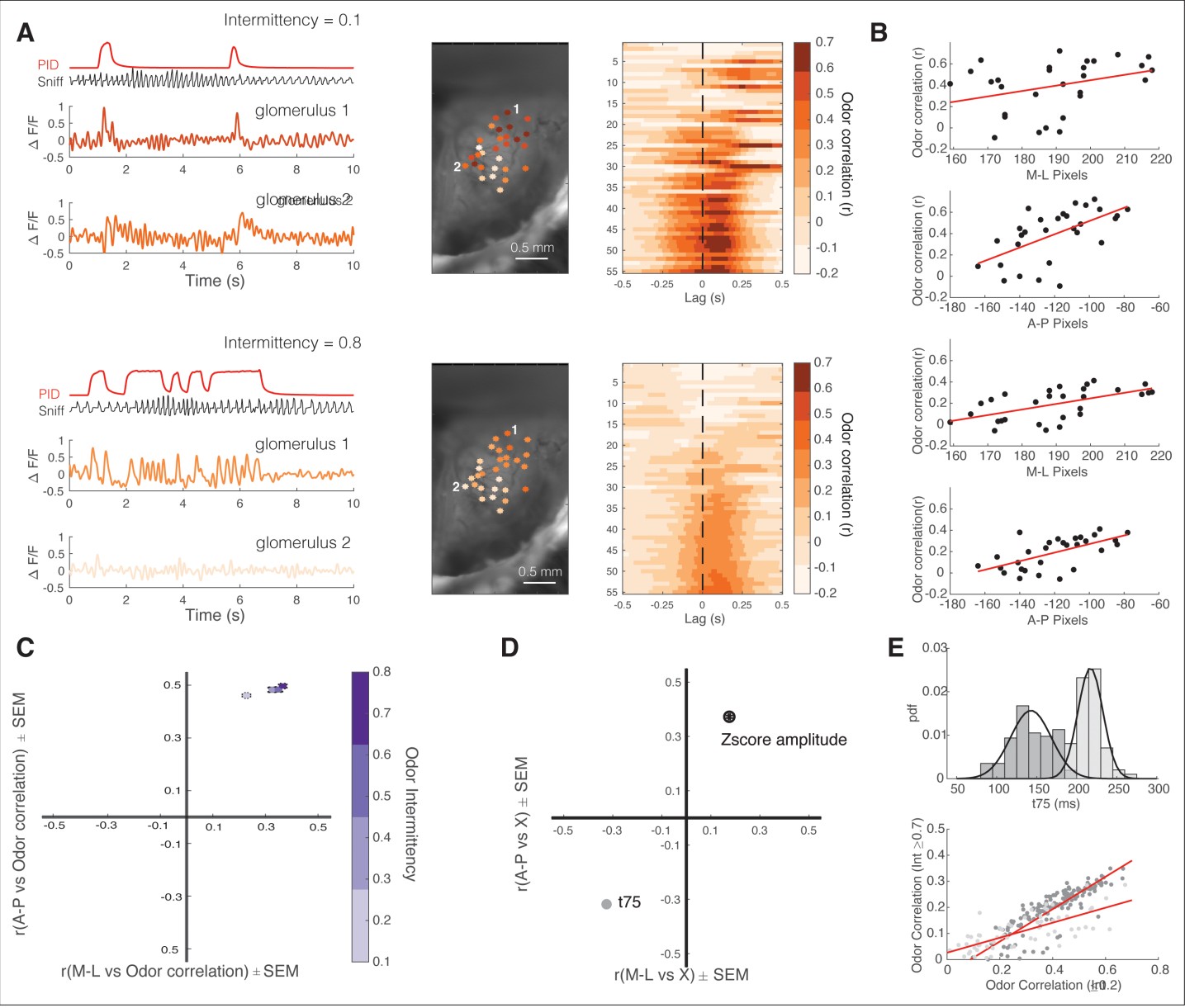

**Figure 4.** Spatial mapping of glomerular response properties across intermittency. (**A**) Example low intermittency trial (*top*) and a high intermittency trial (*bottom*). Photoionization detector (PID) trace (red), sniff trace (black), and deconvolved ΔF/F traces of two example glomeruli (*left,* color coded based on odor correlation color bar, *right*). Example spatial maps of glomeruli color coded based on glomerular response correlation with odor. Two example glomeruli shown in the *left* traces are labeled, *middle*. Cross-correlation between deconvolved ΔF/F and odor for each glomerulus for example trials, *right*. (**B**) Glomerular odor correlation organized based on glomerulus anterior to posterior and medial to lateral location in the dorsal olfactory bulb (low intermittency example, top two graphs; high intermittency example, bottom two graphs). Low intermittency: M-L r=0.34, A-P r=0.58; high intermittency: M-L r=0.58, A-P r=0.62. (**C**) Correlation coefficient of glomerular odor correlation in each dimension (based on graphs in B for all trials). Trial averages are separated by odor intermittency value (colorbar). M-L:$\mu_{int0.1-0.2}$=0.23, $\mu_{int0.3-0.4}$=0.33, $\mu_{int0.5-0.6}$=0.35, $\mu_{int0.7-0.8}$=0.37; A-P: $\mu_{int0.1-0.2}$=0.46, $\mu_{int0.3-0.4}$=0.48, $\mu_{int0.5-0.6}$=0.48, $\mu_{int0.7-0.8}$=0.50. (**D**) Spatial odor map (z-score amplitude, open circle) and spatiotemporal odor map (T75, gray) (for methyl valerate). M-L: $\mu_{z-score(Amplitude)}$=0.18, $\mu_{T75}$=-0.33; A-P: $\mu_{z-score(Amplitude)}$=0.37, $\mu_{T75}$=-0.32. (**E**) Probability density function of T75 for all glomeruli and Gaussian curve fits for fast responding glomeruli cluster (dark gray) and slow responding glomeruli cluster (light gray) (*top*). Glomerulus odor correlation on trials with intermittency ≥0.7 vs glomerulus odor correlation on trials with intermittency ≤0.2. Fast responding glomeruli (low T75): y=0.65x–0.05, $r^2$=0.77, p<0.0001. Slow responding glomeruli (high T75): y=0.29x+0.026, $r^2$=0.53, p<0.0001 (n=244 glomeruli).

The online version of this article includes the following figure supplement(s) for figure 4:

**Figure supplement 1.** Relationship between odor correlation and spatial location, response amplitude, and t75.

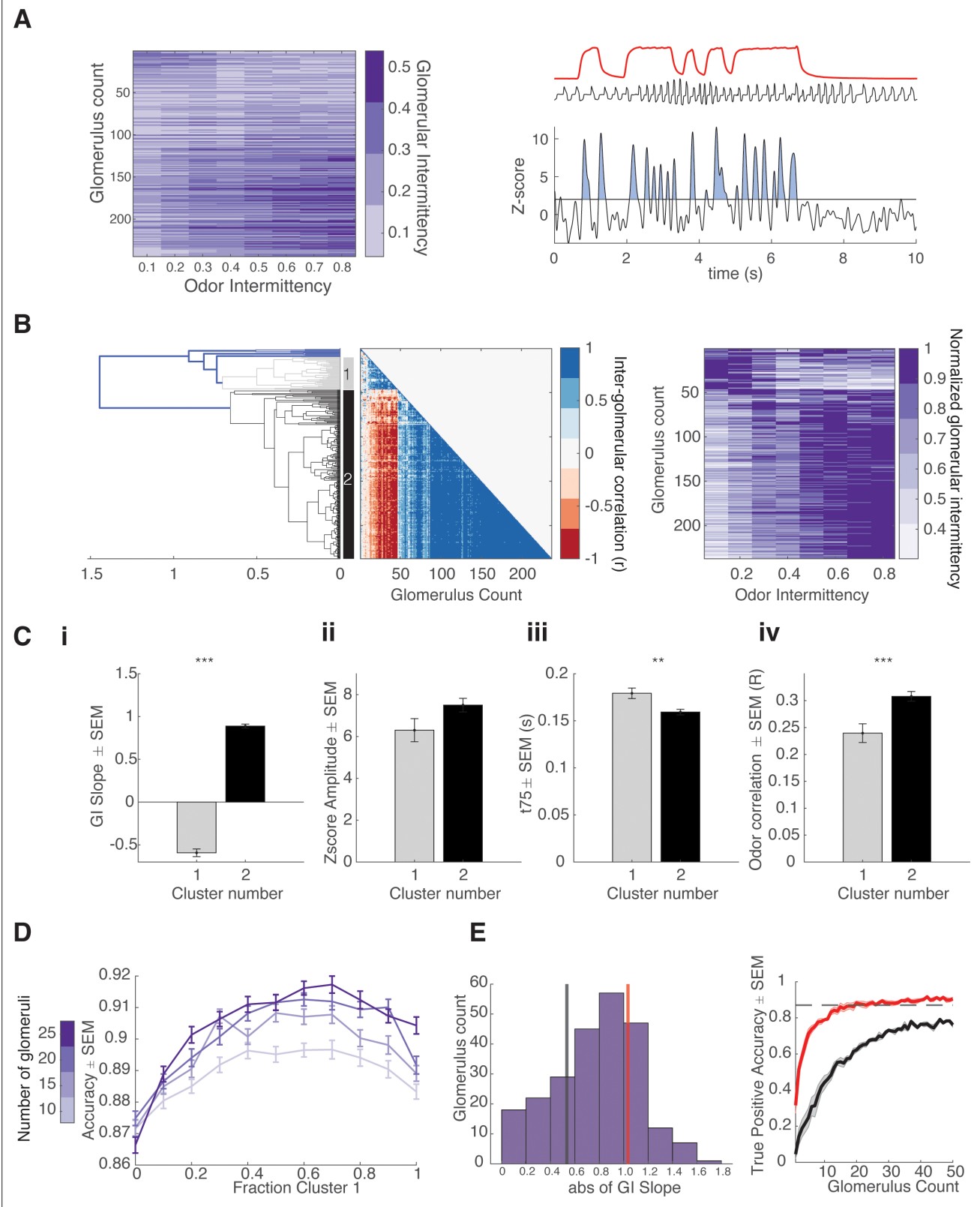

**Figure 5.** Intermittency encoded in olfactory sensory neuron (OSN) glomerular subpopulations. (**A**) Color maps of glomerular intermittency binned by trial odor intermittency. Glomeruli are sorted by their glomerular intermittency (GI) slope (glomerular intermittency vs odor intermittency). Left: Example photoionization detector (PID) odor trace (red), raw sniff trace (black), and z-scored trace from one glomerulus. Horizontal line at y=2 indicates the threshold for glomerular intermittency quantification. (**B**) Left: Dendrogram for hierarchical cluster analysis. Gray indicates cluster 1 (37 glomeruli)

*Figure 5 continued on next page*

*Figure 5 continued*

and black indicates cluster 2 (191 glomeruli). Middle: Inter-glomerular correlation matrix. Colorbar corresponds to correlation coefficient (r) between two glomeruli (glomerular intermittency vs odor intermittency). Right: Colormap of normalized glomerular intermittency (normalized to individual glomerular maximum for clearer visualization of two clusters). Rows sorted by hierarchical clustering. (**C**) (i) Average slope of glomerular intermittency vs odor intermittency of cluster 1 and cluster 2 ($\mu_{cluster1}$=−0.59±0.044, $\mu_{cluster2}$=0.89±0.024). (ii) Average z-score response amplitude for glomeruli in cluster 1 and cluster 2 ($\mu_{cluster1}$=6.3±0.55, $\mu_{cluster2}$=7.5±0.33). (iii) Average T75 for glomeruli in cluster 1 and cluster 2 ($\mu_{cluster1}$=179.2±5.6 ms, $\mu_{cluster2}$=159.3±3 ms). (iv) Average correlation between glomerular deconvolved ΔF/F traces and PID reading for glomeruli in cluster 1 and cluster 2 ($\mu_{cluster1}$=0.24±0.017, $\mu_{cluster2}$=0.31±0.009). (**D**) Accuracy of linear classifier trained using 10, 15, 20, and 25 glomeruli (colorbar) at varying fractions of cluster 1 and cluster 2 glomeruli. (**E**) Left: Histogram of abs(GI slope) for all glomeruli. Black line indicates the bottom 25th percentile (0.53) and red line indicates the top 25th percentile (1.03). Right: True positive accuracy (CS+ predicted as CS+) of linear classifier trained on 0–50 glomeruli for glomeruli with the top 25th percentile of GI slopes (red) and the bottom 25th percentile of GI slopes. Dashed line indicates hit rate of animals on behavioral task (0.87).

The online version of this article includes the following figure supplement(s) for figure 5:

**Figure supplement 1.** Example clusters from three different animals.

**Figure supplement 2.** Intermittency encoded in mitral and tufted (M/T) cell glomerular subpopulations.

classifier, we found that the glomeruli in the top 25th percentile of GI slopes predict trial outcome better than those in the bottom 25th percentile of GI slopes. Just 22 glomeruli with high GI slopes are enough to predict trial outcome at the same accuracy as the average animal hit rate (***Figure 5E***). Thus, even a small number of glomeruli have access to enough information to encode trial intermittency and heterogeneity among glomerular responses to stimuli of varying intermittency may be beneficial for intermittency discrimination. Using THY1-GCaMP6f mice, in which GCaMP6f is expressed in output cells of the olfactory bulb, we found similar glomerular populations as well as a consistent ability of these populations to predict trial identity (***Figure 5—figure supplement 2***). This suggests that access to information that can encode intermittency may arise at the level of olfactory input.

## Sniff frequency-dependent glomerular representation of intermittency

While performing the intermittency discrimination task, mice modulate their active sampling behavior. To systematically measure the effect of sniff frequency on glomerular representation of intermittency, we performed a double tracheotomy procedure on anesthetized animals and used an artificial sniffing system to control their nasal airflow (***Cheung et al., 2009***). In this design, nasal airflow is decoupled from tracheal breathing. We tested glomerular responses to odor stimuli with a range of intermittency values at sniff frequencies of 2, 4, 6, and 8 Hz. This range of sniff frequencies represents those observed during rest to those observed during engaged active sampling (***Wesson et al., 2008***). We presented anesthetized animals with two fruit-associated odors, methyl valerate and 2-heptanone, with neutral preference indices in mice (***Fletcher, 2012***; ***Saraiva et al., 2016***). In addition to having different functional groups, these odors also elicit different spatiotemporal response properties in the olfactory bulb (***Figure 6—figure supplement 1A***). Overall, in both OSNs and M/T cell populations, each individual glomerulus represented a small range of GI values, whereas the entire population encoded a much larger range of GI values, representative of the range of stimulus intermittency values (***Figure 6A***, ***Figure 7A***, glomeruli sorted by the GI slope at 2 Hz, ***Figure 6—figure supplement 1B***). Additionally, there is an interaction effect between odor intermittency, sniff frequency, and genotype on GI (***Figure 6—figure supplement 1C***, GLM, GI~intermittency+sniff frequency+genotype, methyl valerate: $p_{3\text{-way\_interaction}}$ <0.0001, $r^2$=0.40; 2-heptanone: $p_{3\text{-way\_interaction}}$ <0.0001, $r^2$=0.59).

Given that intermittency, sniff frequency, and cell type all have a significant effect on glomerular representation of intermittency, we sought to further probe the influence of these factors. To quantify the effect of sniff frequency on recruiting glomeruli that encode intermittency differences, the number of glomeruli that show a significant effect of odor intermittency on GI was determined. When presented with methyl valerate, OSNs show a 13.4% decrease in the number of intermittency-encoding glomeruli from 2 to 8 Hz sniff frequency (***Figure 6B***, one-way ANOVA, 2 Hz = 82.3% intermittency-encoding glomeruli, 8 Hz = 68.9% intermittency-encoding glomeruli), whereas M/T cells show a 25.2% increase in the number of intermittency-encoding glomeruli from 2 to 8 Hz sniff frequency (***Figure 6B***, one-way ANOVA, 2 Hz = 36.7% intermittency-encoding glomeruli, 8 Hz = 61.9% intermittency encoding glomeruli). When presented with 2-heptanone, both OSNs and M/T cells show an increase in the number of intermittency-encoding glomeruli from 2 to 8 Hz sniff frequency, showing a 22.4% and 25.1% increase, respectively (***Figure 7A and B***, OSN 2 Hz = 65.2%, OSN 8 Hz = 87.6%, M/T 2 Hz = 29.5%, M/T 8 Hz =

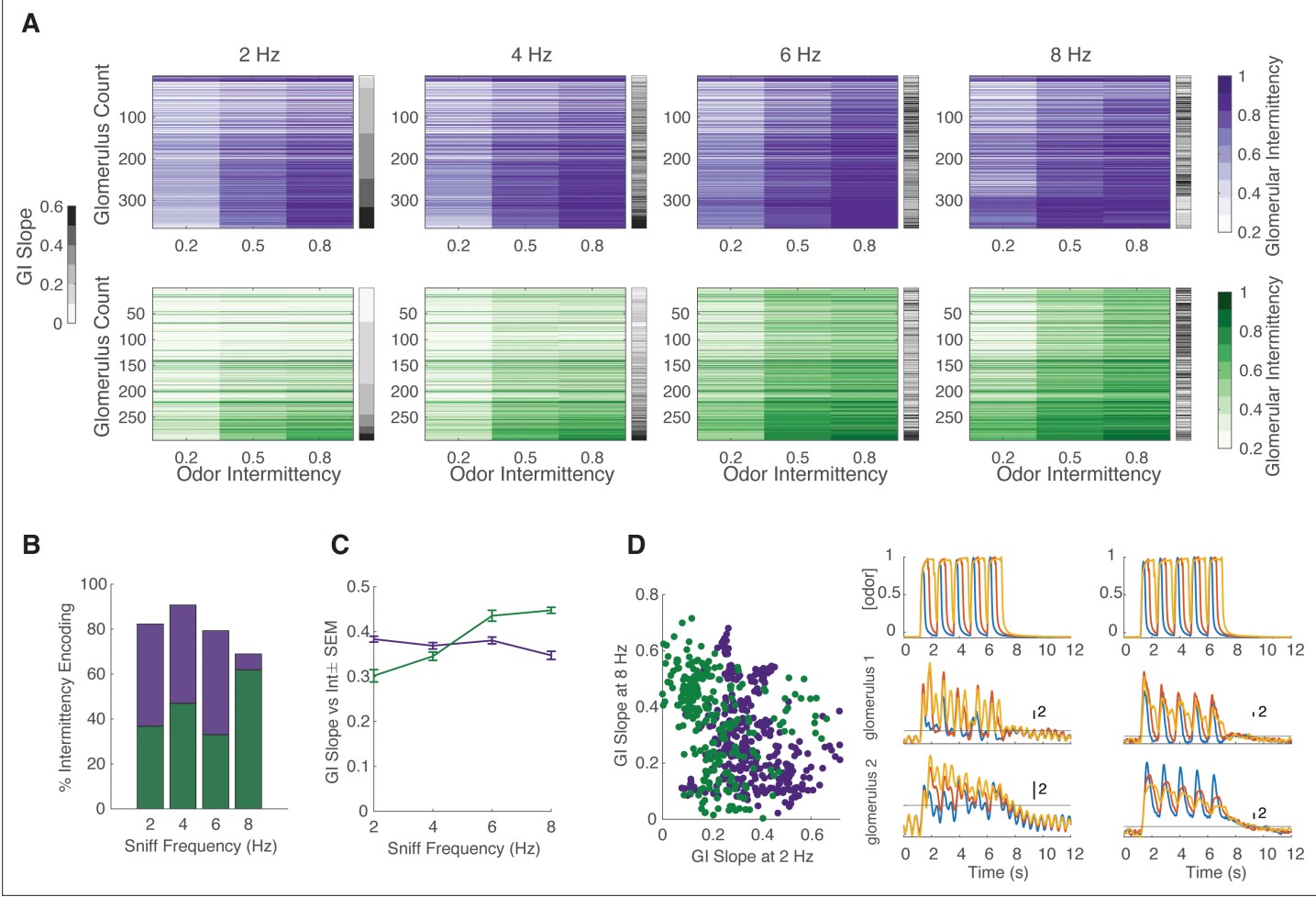

**Figure 6.** Effect of sniff frequency on glomerular representation of intermittency (methyl valerate). For all graphs purple indicates olfactory sensory neurons (OSNs) (OMP-GCaMP6f) and green indicates mitral and tufted (M/T) cells (THY1-GCaMP6f). (**A**) Heatmap of glomerular intermittency (GI) across trials of 0.2, 0.5, and 0.8 odor intermittency values (colored heatmaps). Glomeruli are sorted based on their GI slope at 2 Hz. Gray bars next to heatmaps indicate the GI slope of each individual glomerulus. (**B**) % of intermittency encoding cells across sniff frequencies (OMP n=367 glomeruli, 7 mice; THY n=294 glomeruli, 6 mice). (**C**) GI slope as a function of sniff frequency. (**D**) Left: GI slope at 8 Hz as a function of GI slope at 2 Hz. Right: The top row shows example photoionization detector (PID) readings from square-wave trials with a fixed odor frequency of 0.83 Hz (5 pulses in 6 s) at intermittency values of 0.2 (blue), 0.6 (red), 0.8 (yellow). The first column represents averages from 2 Hz sniff frequency trials and the second column represents averages from 8 Hz sniff frequency trials. The second row shows example z-score deconvolved dF/F traces of a glomerulus with a low GI slope at 2 sniff frequency Hz and a high GI slope at 8 sniff frequency Hz. The last row shows example z-score deconvolved dF/F traces of a glomerulus with a high GI slope at 2 Hz and a low GI slope at 8 Hz. Black line at y=2 indicates the threshold for determining intermittency (z-score value of 2).

The online version of this article includes the following figure supplement(s) for figure 6:

**Figure supplement 1.** Additional quantification of the effect of sniff frequency on glomerular representation of intermittency.

**Figure supplement 2.** Effect of sniff frequency on odor correlation, sniff correlation, air response, and odor response.

54.6%). Additionally, previous studies have shown that increases in sniff frequency lead to more diversity in neural responses of M/T cells (*Díaz-Quesada et al., 2018*; *Jordan et al., 2018a*). Similarly, we find that at higher sniff frequencies, glomerular intermittencies are more variable (*Figure 6—figure supplement 1D*, Bartlett's test, p<0.0001). Having shown that glomeruli with a greater change in GI across odor intermittency, GI slope, can better predict trial intermittency, we quantified changes in glomerular GI slope based on sniff frequency. When presented with both methyl valerate and heptanone, M/T cells that significantly encode intermittency also show an increase in their GI slope as sniff frequency increases (*Figure 6C*, *Figure 7C*, Spearman correlation, methyl valerate: r=0.36, p<0.0001, n=294 glomeruli; 2-heptanone: r=0.29, p<0.0001, n=271 glomeruli). These increases in GI slope may in part be due to the increases in odor tracking (i.e. cross-correlation between glomerular responses

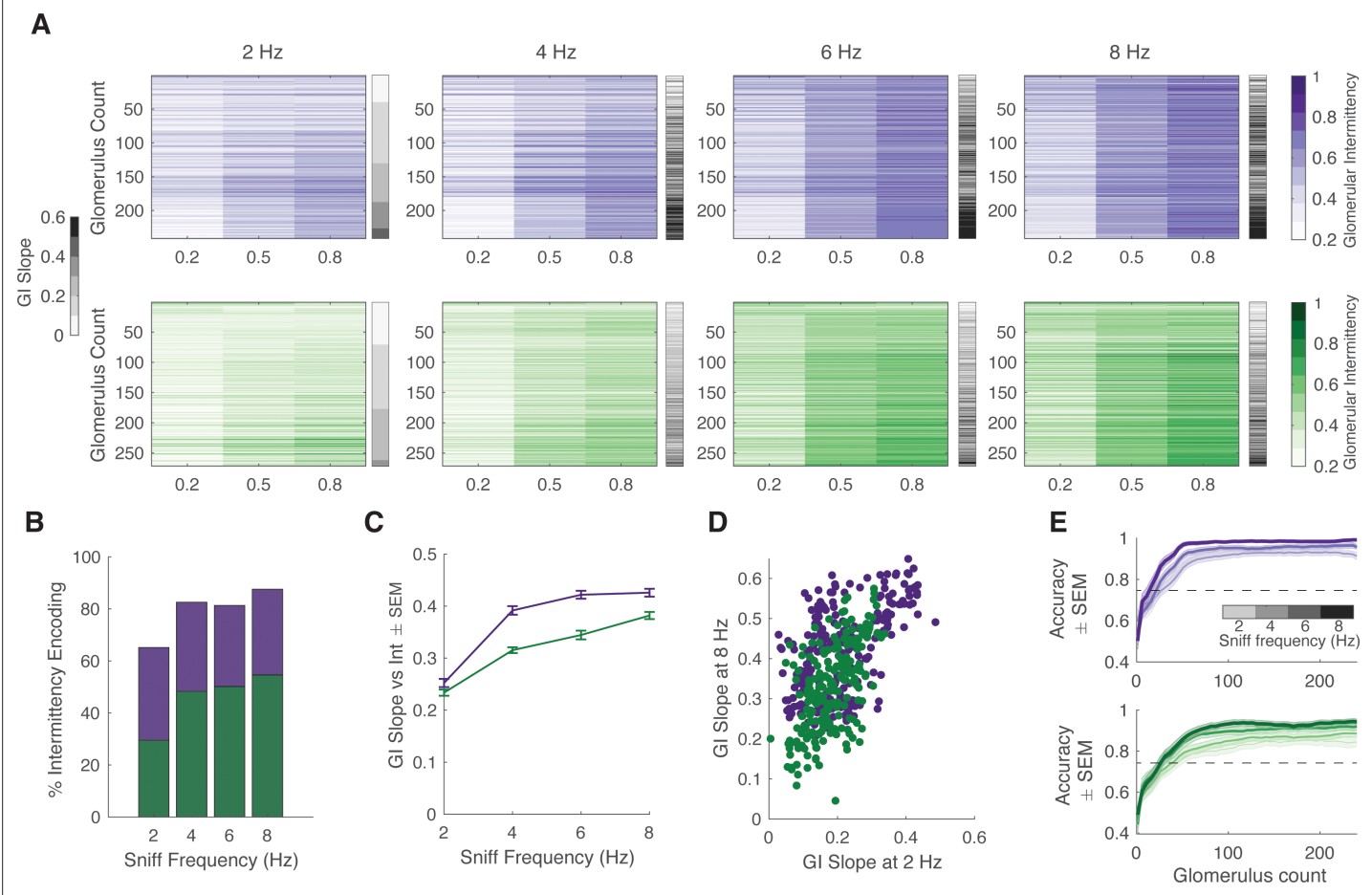

**Figure 7.** Effect of sniff frequency on glomerular representation of intermittency (2-heptanone). For all graphs purple indicates olfactory sensory neurons (OSNs) (OMP-GCaMP6f) and green indicates mitral and tufted (M/T) cells (THY1-GCaMP6f). (**A**) Heatmap of glomerular intermittency across trials of 0.2, 0.5, and 0.8 odor intermittency values (colored heatmaps). Glomeruli are sorted based on their glomerular intermittency (GI) slope at 2 Hz. Gray bars next to heatmaps indicate the GI slope of each individual glomerulus. (**B**) % of intermittency encoding cells across sniff frequencies (OMP n=241 glomeruli, 6 mice; THY n=271 glomeruli, 6 mice). (**C**) GI slope as a function of sniff frequency. (**D**) GI slope at 8 Hz as a function of GI slope at 2 Hz. (**E**) Linear classifier performance (accuracy) over 240 glomeruli when trained on trials of four different sniff frequencies (2, 4, 6, 8 Hz). 60 iterations (20 times threefold) per classifier. Exponential plateau fit, OMP: 2 Hz: plateau = 0.95, Y=0.95–(0.44)*(e$^{-0.04x}$), r$^2$=0.67; 8 Hz: plateau = 0.98, Y=0.98–(0.37)*(e$^{-0.04x}$), r$^2$=0.7; THY: 2 Hz: plateau = 0.85, Y=0.85–(0.4)*(e$^{-0.03x}$), r$^2$=0.54; 8 Hz: plateau = 0.94, Y=0.94–(0.4)*(e$^{-0.04x}$), r$^2$=0.64.

and PID signal), decreases in sniff tracking (i.e. cross-correlation between glomerular responses and sniff pressure signal), and decreases in air responses as sniff frequency increases (*Figure 6—figure supplement 2*). Together this suggests that at higher sniff frequencies, more glomeruli encode intermittency, as well as show a greater range in their representation of intermittency at the level of olfactory bulb output.

We next assessed if the same glomerular populations best represent intermittency at both low and high sniff frequencies. Using GI slope to represent the degree to which glomeruli contribute to intermittency prediction, we found that when presented with methyl valerate, both input and output cells show a negative relationship between GI slope at 8 and at 2 Hz sniff frequency (*Figure 6D*, *left and right,* linear regression, OMP: y=−0.27x+0.38, r$^2$=0.05, p<0.0001; THY: y=−0.40x+0.44, r$^2$=0.11, p<0.0001). This shows that different glomeruli show the greatest change in GI across odor intermittency (GI slope) at 2 and 8 Hz sniff frequency, suggesting that different glomerular populations encode intermittency at low and high sniff frequencies. However, we found that when presented with 2-heptanone, the same glomeruli have the greatest GI slope at both 2 and 8 Hz sniff frequency (*Figure 7D*, linear regression, OMP: y=0.57x+0.29, r$^2$=0.35, p<0.0001; THY: y=0.94x+0.15, r$^2$=0.35, p<0.0001). Together this leads to the understanding that the effect of sniff frequency on the glomerular

population that most strongly encodes intermittency is odor-specific (GLM: GI slope~sniff frequency*odor, p<0.0001, $r^2$=0.14, sniff frequency*odor interaction, p<0.0001).

Although sniff frequency has an effect on both the number of glomeruli that encode intermittency and the strength of this encoding (GI slope), there are only slight differences between the prediction of odor stimulus intermittency based on sniff frequency. We trained a linear classifier to predict odor intermittency value (0.2, 0.5, or 0.8) based on trial GI across sniff frequencies. Using glomeruli responding to methyl valerate, the classifier performs better at low sniff frequencies and plateaus for OSNs at 91% for 2 Hz and 86% for 8 Hz (*Figure 6—figure supplement 1E*, exponential plateau fit). For M/T cells, the classifier plateaus at 92% for 2 Hz and 93% for 8 Hz, showing little difference in performance between trials of different sniff frequencies. Using glomeruli responding to 2-heptanone, the classifier performs better at high sniff frequencies for both OSNs and M/T cells (*Figure 7E*, exponential plateau fit, OSNs: plateaus at 95% for 2 Hz and 99% for 8 Hz, M/Ts: plateaus at 85% for 2 Hz and 94% for 8 Hz). However, for both input and output cells, across all sniff frequencies and odors tested, less than 50 glomeruli are required to exceed a prediction accuracy of 75% (*Figure 7E*, *Figure 6—figure supplement 1E*). Overall, this suggests that although sniff frequency has an effect on prediction of odor intermittency, prediction accuracy is high using glomerular information from trials at all sniff frequencies between 2 and 8 Hz. This finding is congruent with previous studies showing that although sniff frequency alters olfactory bulb response properties, it does not influence performance on odor discrimination tasks (*Jordan et al., 2018b*).

## Discussion

Mammals are likely often required to rely on and navigate within a highly dynamic and complex odor plume environment in order to find food sources, locate mates, and avoid predators (*Vergassola et al., 2007*; *Reddy et al., 2022*). Although it is well established that mammals are skilled at navigating within these complex environments, the properties of odor plumes that they use for source localization remain largely unknown (*Baker et al., 2018*). Here, we show that mice can discriminate odor intermittency, a temporal odor plume property that varies with distance from the odor source, and that early olfactory processing encodes intermittency. We demonstrate that active sampling patterns may affect intermittency discrimination and sniff frequency influences glomerular representation of intermittency. Additionally, we have shown that glomeruli encode information that enables reliable discrimination between odor plume samples based on intermittency. Overall, these findings suggest that intermittency can be used to inform odor-guided navigation in mice.

We found that mouse performance on the intermittency discrimination task is not affected by the odor used or frequency of odor whiffs, but is affected by the concentration gain. This shows that intermittency is a temporal property of odor plumes that can be detected independently from other temporal properties, such as whiff frequency. This distinction may be important if different temporal properties provide at least partially independent information about location within the odor plume, as suggested by *Jayaram et al., 2022*. Other temporal properties that may indicate distance from and composition of an odor source are odor whiff frequency and the temporal correlation of fluctuating odors, both of which mice are capable of detecting (*Hopfield, 1991*; *Schmuker et al., 2016*; *Ackels et al., 2021*; *Dasgupta et al., 2022*). It is possible that these temporal properties are either used independently or in concert during odor navigation. Additionally, we show that concentration gain has an effect on intermittency discrimination, suggesting that mice are in part using odor concentration integration for intermittency discrimination. While further work needs to be done to explore discrimination of odor stimuli based on odor integration, a plethora of work suggests that rodents can discriminate odor duration and intensity both at the neural and behavioral levels (*Rubin and Katz, 1999*; *Rospars et al., 2000*; *Spors and Grinvald, 2002*; *Li et al., 2014*; *Wojcik and Sirotin, 2014*; *Sirotin et al., 2015*; *Li et al., 2020*). In some odor plumes, odor concentration and odor intermittency both increase as distance from the odor source decreases, indicating that the integral of measured odor would also increase (*Connor et al., 2018*). It is possible that odor intermittency and odor integration might inform odor source localization in a partially dependent manner, where both statistical properties provide information on location within the plume.

We found that glomeruli show heterogeneous responses to odor plume stimuli across a range of intermittency values. We primarily found two subsets of glomeruli with inverse representations of intermittency and select glomeruli within both populations best represent changes in intermittency.

Our results are consistent with previous findings showing that a subset of glomeruli track odor plume dynamics, and that the degree of odor tracking depends on odor plume statistics (*Lewis et al., 2021*). However, our findings contradict those suggesting that fluctuating odors are linearly processed by M/T cells (*Gupta et al., 2015*). This discrepancy may be due to our use of stimuli across a range of intermittencies and resulting non-linearities may only arise in response to plume samples of certain odor plume statistics not previously tested. Biophysical diversity in the mammalian olfactory bulb allows for increased information encoding (*Padmanabhan and Urban, 2010*; *Tripathy et al., 2013*), which may be particularly relevant during odor plume navigation where the odor plume property that is most salient for source localization may change along an animal's trajectory within the plume (*Rigolli et al., 2021*; *Jayaram et al., 2022*). Thus, the heterogeneity we observe among glomeruli may yield specialized populations, important for feature selection of different statistical properties within the odor plume. Ultimately, these populations could be advantageous for efficient information encoding during odor plume navigation. Our results suggest glomeruli have varying lag times in their odor correlation and if future studies indicate a response synchronicity within these glomerular clusters, this concerted activity could be important in robustly encoding intermittency (*Gill et al., 2020*). We show that both populations can predict intermittency and show distinct response properties (*Figure 5*). Cluster 2 glomeruli, which show higher glomerular-response correlations may not only be effective at encoding intermittency but may also convey information about odor whiff timing or frequency; both of which provide additional information about an animal's distance to the odor source (*Crimaldi and Koseff, 2001*; *Celani et al., 2014*).

There are several properties of glomerular responses that can be altered by sniff frequency which can influence intermittency representation. Some of the properties that we have considered that may cause an increase in GI representation are a reduction of baseline air responses (the threshold for GI is dependent on the background response), an increase in response amplitude (so that it exceeds the threshold), an increase in the number of responses per unit time, or a decrease in adaptation (preventing sustained response from dropping below the threshold). We find that the average glomerular odor-response amplitude does not change with sniff frequency, but the air response amplitude decreases as sniff frequency increases. At low sniff frequencies, responses are more highly correlated with the sniff trace, whereas at higher sniff frequencies, responses are more highly correlated with the odor trace. This suggests that at higher sniff frequencies, odor responses are more representative of the stimulus due to more frequent sampling (*Figure 6*). Additionally, although high-frequency sniffing attenuates odor responses (*Verhagen et al., 2007*; *Wachowiak et al., 2009*), GI on average increases with sniff frequency, suggesting that glomerular responses do not fall below the threshold for GI calculation due to adaptation. Using these findings, we reason that high sniff frequency in part changes intermittency representation by decreasing background air responses and more accurately representing the odor stimulus. These changes alone provide a reasonable understanding of why we observe an increase in M/T cell representation of changes in intermittency at high sniff frequencies.

When exploring the role of intermittency in mouse odor-based navigation, two main questions to address are: Is it feasible for mice to detect changes in odor intermittency? If so, are they using intermittency for navigation? We address the former by using a behavioral assay to determine that mice can discriminate odor intermittency and that the olfactory bulb can support this discrimination. The results of our work provide motivation to further study how mice may use intermittency when navigating complex odor environments. Future work focusing on shifts in navigation strategies based on changes in odor plume intermittency will further elucidate how mice use this property for source localization and enable more informed modeling approaches to the use of intermittency. Recently, head-mounted sensors have been implemented to measure odor concentration in real time during plume navigation in a laboratory arena (*Tariq et al., 2021*). Such techniques provide important advancements that can be used to understand the correlation between changes in odor plume statistics experienced by a navigating animal and the behavioral decisions made along its trajectory. However, a limitation in studying rodent navigation in the laboratory environment is the ability to recapitulate naturalistic odor environments where more complex navigation strategies may be needed. To address this, virtual odor environments have begun to be implemented (*Baker et al., 2018*; *Radvansky and Dombeck, 2018*; *Fischler-Ruiz et al., 2021*). The use of virtual odor plumes presents an opportunity to manipulate the odor environment in a systematic manner and expose differences in navigation strategies when for example intermittency changes with distance from the odor source versus when it does not (*Jayaram*

*et al., 2022*). Measuring if mice can detect within-stimulus increases or decreases in odor plume intermittency will also help elucidate the timescale over which mice can detect intermittency differences.

Our data show that mice can discriminate odor intermittency and that early processing in the mouse olfactory system encodes intermittency. Intermittency influences and may be critical for odor navigation in invertebrates. Here, we take the first steps in showing that intermittency may be used by rodents for odor plume navigation and provide further support that timing-based properties can be used for source localization.

## Methods

### Olfactometer design

Filtered high purity nitrogen (Airgas, NI ISP300, <0.1 ppm total hydrocarbons, $H_2O$, and $O_2$ contaminants) was carbon filtered and passed through PFA vials (Savillex 200-30-12) containing odor (2% methyl valerate in mineral oil, Sigma-Aldrich product #1489977; 2% 2-heptanone in mineral oil, Sigma-Aldrich product # 537683; stored in the dark under nitrogen at room temperature). Additionally, the carbon-filtered nitrogen line was split before passing through the odor and is directed to an empty PFA vial used for counterbalancing odor flow (*Figure 1B*). Both odor delivery and counterbalanced nitrogen delivery are controlled by independent EVP Series Clippard Proportional Valves and an EVPD-2 valve driver (Clippard Instrument Laboratory, Inc, Cincinnati, OH, USA). Proportional valves were calibrated so as to ensure that final combined nitrogen and odor flow is maintained at 50 mL/min. Combined nitrogen and odor flow was confirmed using a flow meter (Omron Electronic Components Product # D6F-P0010A1). Combined nitrogen and odor flow is injected orthogonally to and diluted using clean air (Airgas, AI UZ300, ultra-zero grade, <1 ppm total hydrocarbons, $CO_2$, and CO contaminants) that is carbon filtered and passed through a mass flow controller so that it is maintained at 200 mL/ min. All connections within the olfactometer design were made using 1/8" OD Teflon tubing (4 mm ID, 8 mm OD). Odor delivery is confirmed using a mini PID (200B miniPID, Aurora Scientific). Suction flow through the PID was fixed at 90 mL/min through a flow meter (Cole Parmer PMR1-010977). Final airflow post PID suction is 160 mL/min and the odor dilution is delivered through a Teflon nose cone. Final odor delivery consists of a 0–20% airflow dilution of a 2% liquid odor dilution in mineral oil.

### Olfactory stimuli

#### Plume data

Odor plume data was collected in the lab of Dr. John Crimaldi using a flow chamber compatible with planar laser-induced fluorescence according to the specifications indicated by *Connor et al., 2018*. Digital mapping of this data has been made available on the DataDryad database (DataDryad, https://doi.org/10.5061/dryad.zgmsbcc71). Odor plume traces (6 s each) within the digital odor plume were made at varying distances directly in line with the odor source (60, 120, and 240 y pixel coordinates from the release point, x=245 and y=0, out of 495×495 pixels) to obtain stimuli of varying intermittency values. Samples were normalized so that all traces reached the same maximum concentration so as to eliminate effects of maximum concentration differences dependent on distance from the odor source. Stimuli were delivered at two gain levels, a gain of 1 and a gain of 0.5. For the latter, the sample trace was scaled so that the maximum concentration was half of that delivered for a gain of 1. Intermittency is the probability that the odor concentration exceeds a threshold at a location of an odor plume. Intermittency γ values of each sample were calculated according to the following equation, where C is the concentration trace for a given stimulus and $C_0$ is the source concentration:

$$\gamma = \mathrm{prob}\left[C \geq 0.1C_0\right]$$

#### Naturalistic stimuli

Eight unique odor traces were extracted from the digital odor plume with an intermittency value ≤0.15 and four unique stimuli were created for each interval of 0.1 for intermittency values between 0.2 and 0.8 (e.g. four unique stimuli with intermittency values 0.2≤γ≤0.3).

## Binary naturalistic stimuli

Binary naturalistic stimuli were generated using the same time-concentration traces as the naturalistic stimuli. Binary naturalistic stimuli were binarized using a threshold of $0.1C_0$ so that all $C \geq 0.1C_0$ reached the maximum concentration. In this set of stimuli, there were no intermediate concentration changes (i.e. odor was either at maximum concentration, 20% of final airflow dilution using 2% liquid odor in mineral oil, or off).

## Square-wave stimuli

Square-wave stimuli were generated using the Square() function in MATLAB (MATLAB 2021b, Math-Works, MA, USA) with periods between $1/3\pi$ and $8/3\pi$ (pulse repetitions between 1 and 8 pulses in 6 s at intervals of 1 pulse) and with duty cycles between 10 and 80 at intervals of 10.

## Optical imaging system and experimental setup

Olfactory bulbs of awake head-fixed animals were imaged using wide-field fluorescent microscopy. A high-power LED 470 nm (Thorlabs, Newark, NJ, USA) stimulation driven by a T-Cube LED Driver (LEDD1B, Thorlabs, Newark, NJ, USA) was used for the duration of the 9 s trial (but remained off during the inter-trial interval to avoid photobleaching effects). Imaging was collected using a RedShirtImaging NeuroCCD256 optical imaging system. The epifluorescence macroscope used is a custom-made tandem-lens type with a 135 mm F/3.5 Nikon objective lens and 85 mm F/4 Nikon imaging lens, yielding a ×1.59 magnification and 4.2 mm field of view. The fluorescence filter set is BL P01-514 (excitation filter), LP515 (dichroic), and LP530 (emission filter; Semrock, Lake Forest, IL, USA). Data was collected using Neuroplex Software (RedshirtImaging) and converted into MATLAB-compatible files for further analysis.

Animals were head-fixed to a custom stationary bar made to fit the stainless-steel head-post over a freely-rotating 20 cm diameter wheel moving on a spinning axis. The behavioral setup was equipped with an automated imaging-compatible Go/No-Go task setup implemented using custom LabView software. A rotary encoder (Broadcom/Avago HEDS-5500/5600 series) was attached to the axis of the wheel to measure running speed. An infrared beam break sensing lick-spout allows for lick-based reward delivery and lick-counting. Additionally, a pressure sensor (Amphenol 0.25 INCH-D-4V) is inserted into the nose cone to measure sniffing during the behavioral task.

For pupil tracking, an infrared CMOS camera (Basler, acA1920) was positioned in front of the animal along with an 850 nm LED (M850L3, Thorlabs) and an 850±8 nm bandpass filter (FB850-40, Thorlabs), to illuminate the eye. Image acquisition from the pupil camera was synchronized with the start of each trial. Custom software code written in LabView (National Instruments) controlled image acquisition, storage, and data analysis. Images were acquired at 30 Hz for the duration of the trial and analyzed in real time to extract the pupil diameter. Each frame of the image series was passed through an edge detection algorithm developed using the vision development module in LabView. A region-of-interest in the shape of an annulus was drawn over the pupil with an inner circle near the center of the eye and the outer circle extending past the edge of the pupil with enough room to allow for dilations and constrictions. The edge detection algorithm identified dark to light transitions points starting from the inner circle to the outer circle. The pupil diameter was calculated by fitting a circle using the detected edges. The threshold level for edge detection and number of transitions points to be identified were adjusted to get the best fit for each mouse.

## Artificial sniffing system

Components for the artificial sniffing system used a mounted 5 mL glass syringe piston (Air-Tite, 7.140-33) coupled via a custom 3D-printed connector to a linear solenoid actuator (Soft Shift Part# 192907-023) under the control of a voltage-driven command (Canfield Connector B950 Series) by custom LabView software. The driven movement of the actuator allows for gradual push and pull of air through the syringe. The inlet end of the syringe was attached to tubing connecting to the nasopharyngeal cavity of the tracheotomized mouse. This design is according to the specifications described in *Cheung et al., 2009*. Sniff traces were obtained from the Wachowiak lab (previous published traces in *Cheung et al., 2009*) and resampled to produce traces at sniff frequencies of 2, 4, 6, and 8 Hz.

## Mice

For behavioral experiments, seven OMP-GCaMP6f mice (5 males, 3 females; generated by crossing *OMP-Cre* [Jax Stock #006668; B6;129P2-*Omp^{tm4(cre)Mom}*/MomJ] with *GCaMP6f* floxed transgenic

mice [Jax Stock #024105; B6;129S-Gt(ROSA)*26Sor*$^{tm95.1(CAG-GCaMP6f)Hze}$/J]) and six THY1-GCaMP6f mice (2 males, 4 females; Jax Stock #024339; C57BL/6J-Tg(Thy1-GCaMP6f)GP5.11Dkim/J) aged 10–12 weeks were used. Mice were housed up to 3 per cage under a 12–12 hr reverse light-dark cycle. The experimental design flow was carried out as follows: animals underwent the head-post surgical procedure, 48 hr after head-post-surgery mice were water regulated and handled for 5 days, mice were acclimated to head-restraint for 3 days, mice were trained on the Go/No-Go task (mean ± SD: 3.7 ± 1.8 days), mice were tested on the Go/No-Go task until meeting criteria (mean ± SD: 7.2 ± 6.4 days), water regulation was temporarily suspended 48 hr prior to surgery, mice underwent the optical window procedure, 48 hr after optical window procedure mice were water regulated and tested on the Go/No-Go task while performing calcium imaging on the olfactory bulb (3.7 ± 2.3 days per condition).

For anesthetized imaging, 13 OMP-GCaMP6f mice (heptanone: 3 males, 3 females; methyl valerate: 4 males, 3 females; generated by crossing *OMP-Cre* [Jax Stock #006668] with *GCaMP6f* floxed transgenic mice [Jax Stock #024105]) aged 10–12 weeks and 12 THY1-GCaMP6f mice (heptanone: 3 males, 3 females; methyl valerate: 3 males, 3 females; Jax Stock #024339) were used. Mice were housed up to 3 per cage under a 12–12 hr reverse light-dark cycle.

### Primer sequences

> FL-GCAMP6F: Common Reverse: CCGAAAATCTGTGGGAAGTC; Wild Type Forward: AAGG GAGCTGCAGTGGAGTA; Mutant Forward: ACGAGTCGGATCTCCCTTTG.
> OMP-CRE: Wild Type Forward: AGTTCGATCACTGGAACGTG; Wild Type Reverse: CCCA AAAGGCCTCTACAGTCT; Mutant Forward: TAGTGAAACAGGGGCAATGG; Mutant Reverse: AGACTGCCTTGGGAAAAGCG.
> THY1-GCAMP6F: Mutant Forward: AAAGAGAGGGGCTGAGGTATTC; Mutant Reverse: CTCG AGATCCTCTAGGTGCC.

## Surgical procedures

### Head-post procedure

Head-post implant procedure was performed as outlined in *Baker et al., 2019*. Animals were anesthetized using isoflurane (4% for induction, 1.5% for maintenance) and monitored by testing pedal reflex withdrawal. Core body temperature was monitored using a rectal thermometer coupled to a thermostatically controlled heating pad to maintain a temperature of 37°C. Carprofen (5 mg/kg, s.c.) and buprenorphine (50 μg/kg, i.m.) were administered prior to surgery. The head of the isoflurane-anesthetized mouse was shaved, scrubbed with betadine followed by alcohol, then secured in a stereotaxic head holder. The skin caudal to Bregma was retracted and a 9×40×1.5 mm aluminum plate was cemented to the skull using MetaBond (Parkell C & B Metabond Quick Self-Curing Cement System).

### Optical window procedure

Thinned-skull dorsal olfactory bulb optical window surgery was performed as outlined in *Baker et al., 2019*. Animals were anesthetized, monitored, and provided analgesics as described in the head-post procedure. Mice received supplemental carprofen 24 hr post surgery. Animals were placed in a stereotaxic holder, and the animals were prepared using aseptic procedures. For exposure of the dorsal olfactory bulb, the skin was removed, and the underlying bone was thinned using a dental drill. A thin layer of cyanoacrylate was applied to the dorsal window.

### Tracheotomy procedure for anesthetized imaging

No earlier than 48 hr after optical window instillation, animals were anesthetized with ketamine:dexmedetomidine (100:0.5 mg/kg, i.p., 25% original dose booster). Additionally, animals were administered atropine (0.03 mg/kg, i.p.). The skin of the neck was shaved and scrubbed using betadine followed by alcohol. An incision in the skin was made and muscle bundles overlying the trachea were separated. An incision was made in caudal end of the trachea and sterilized polyethylene tubing (0.86 mm ID, 1.27 mm OD) was installed and directed toward the lungs. A knot was tied tightly around

the trachea and tubing (to prevent flow of any fluid into the trachea) with suture thread (Surgical Specialties #SP102). The same was done at the rostral end of the trachea for the insertion of a nasopharyngeal tube for breathing-independent orthonasal odor presentation. The midline incision was closed by sutures.

## Go/No-Go behavior

### Acclimation

Forty-eight hours after recovery from head-post surgery, animals underwent 5 days of adaptation to experimenter handling and concurrently were started on a regimen of water regulation (access to 1 mL of water per animal each day in their home cage). Animal body weight was maintained at 85% of original body weight. Following 5 days of handling, animals were acclimated to head-restraint over a freely rotating wheel for 3 days. When animals were head-fixed over this wheel, they could run freely.

### CS+ training

Following head fixation habituation, animals were trained to lick in response to the CS+ ($0.2 \leq$ intermittency<0.9). At the beginning of each trial, animals were presented with a 500 ms 2 kHz tone, followed by a 1.5 s delay before odor presentation. Stimulus was presented for 6 s followed by a 500 ms 6 kHz tone indicating the beginning of a 1.5 s decision period. If mice licked before the decision period, the trial still continued, however anticipatory licking was recorded. If mice licked during the decision period, they received a water reward. One mouse out of the eight OMP-GCaMP6f mice was unable to acquire the lick training task and was removed from the study. One mouse out of the seven THY1-GCaMP6f mice was unable to acquire the lick training task and was removed from the study. Once animals successfully licked for >85% of CS+ trials, they were moved onto training on the complete Go/No-Go task. On average, animals took 2.08 ± 0.29 days to reach 85% on CS+ training.

### Complete Go/No-Go task

For the complete Go/No-Go task, animals were trained to lick for stimuli with intermittency values ≥0.2 and withhold licking for stimuli with intermittency values ≤0.15 (CS-). If mice licked during the decision period in response to a CS+, they received a water reward. If mice licked during the decision period in response to a CS-, they received a punishment in the form of an increased inter-trial interval (increasing from 7 to 14 s). For each session 50% of trials are CS+ and 50% of trials are CS- following an initial 8 high intermittency CS+ trials used to engage the animal in the task. On a given session, animals were trained or tested on 64 trials (8 trials of intermittency >0.6 followed by a random presentation of 28 CS+ trials and 28 CS- trials). Additionally, trials of a gain of 0.5 and a gain of 1 are interwoven randomly during the session with each unique stimulus being presented at both a gain of 0.5 and 1. Thus, after the initial engagement trials, animals are presented with a total of 28 trials at a gain of 0.5 and 28 trials at a gain of 1. Animals were trained using a set of naturalistic stimuli. Animals met criteria when they reached a hit rate >75% and a false alarm rate <25% for 2 consecutive days. On average, animals took 3.7 ± 1.8 days to reach criteria on the complete Go/No-Go training set. Once animals met criteria on the training set, they were tested using two sets of naturalistic, binary naturalistic, and square-wave stimuli each. The order in which animals were tested on each stimulus set type was randomly permuted (e.g. some animals were first tested on naturalistic and once criteria was met, they were moved onto square wave, and then lastly onto binary naturalistic. Other animals were started on binary naturalistic and once criteria was met, they were moved onto square-wave, and then lastly onto naturalistic).

All animals are trained and tested using 2% methyl valerate in mineral oil (Sigma-Aldrich product #1489977). After completing the entire behavioral paradigm (testing using naturalistic, binary naturalistic, and square-wave stimulus sets) using 2% methyl valerate, animals are tested on the binary naturalistic condition using 2% 2-heptanone in mineral oil (Sigma-Aldrich product #537683). Thus, within a session, all CS+ and CS- are a single odor, the distinguishing property between CS+ and CS- is their intermittency value.

## Awake wide-field calcium imaging

After being tested on the 2-heptanone control condition, animals underwent an optical window procedure and were allowed a minimum 48 hr recovery period. After the recovery period, animals were tested on the binary naturalistic and square-wave stimulus sets using methyl valerate. The order in which animals were tested on each stimulus set type was randomly permuted. During this testing period, the dorsal olfactory bulbs of these animals were imaged according to the optical imaging protocol described in the 'Optical imaging system' section.

## Anesthetized wide-field calcium imaging

Animals remained anesthetized post tracheotomy and immediately prepared for calcium imaging. They were placed on a heating pad directly under a camera with their nose directly in front of an odor tube. The nasopharyngeal tracheal tube was connected to the previously described actuator-controlled artificial sniffing system while allowing the animal to freely breath through the lung-directed tracheal tube. Core body temperature was monitored and maintained throughout the procedure. Breathing was monitored by eye for signs of distress. Warmed saline was administered for hydration after 4 hr. Imaging lasted no more than 8 hr. Anesthetic maintenance was monitored based on the pedal withdrawal reflex and anesthesia boosters were administered as necessary (ketamine:dexmedetomidine, 100:0.5 mg/kg, i.p., 25% original dose booster). An atropine booster was administered every 2 hr after first administration (0.3 mg/kg, i.p.) Animals were euthanized immediately after imaging.

## Quantification and statistical analyses

All behavioral and imaging data was converted into a MATLAB-compatible format. All quantification and statistical analysis were carried out using custom-written MATLAB scripts.

### Statistics

For all statistical analyses, one asterisk denotes $p<0.05$, two asterisks denote $p<0.01$, and three asterisks denote $p<0.001$. In all cases $p<0.05$ was used to determine significance unless otherwise stated. On all graphs, unless otherwise stated, the error bars indicate standard error (SEM). For all box and whisker plots, the center line indicates the group median and the limits of the box correspond to the upper 0.75 quantile and lower 0.25 quantile. The ends of the whiskers correspond to 1.5*interquartile range from either the top or bottom of the box. Outliers are indicated as points that lie beyond the whiskers.

In order to execute statistical tests, the following MATLAB functions were used: *fitlme* (mixed effects models,), *fitlm* or *corr* (linear regression models), *xcorr* (cross-correlation analyses), *linkage* and *clust* (cluster analysis), and *fitdiscr* and *predict* (linear discriminant analysis).

### Go/No-Go behavior

For each session, animal performance was calculated starting with the first trial following four hit trials to ensure animal engagement in the task. For each session, hit rate was calculated as:

$$HR = \frac{Correct\ CS+}{total\ CS+}$$

with correct CS+ trials being CS+ trials where the animal licked during the decision period. For each session false alarm rate was calculated as:

$$FA = \frac{Incorrect\ CS-}{total\ CS-}$$

with incorrect CS- trials being CS- trials where the animal licked during the decision period. Total animal performance for each session was calculated as: performance = HR-FA. A mixed effects model (hit rate~stimulus type+gain+odor+odor intermittency) controlling for a random effect of animal identity was implemented using MATLAB function fitlme() to determine the effect of various independent variables on animal performance on the task.

As previously described, sniffing was measured in real time using a pressure sensor during task performance. Sniff peaks were identified by peaks in the sniff trace (where positive deflections

indicate inhalation) and sniff onsets were identified as the point at which the sniff trace changes sign prior to the peak. Inhalation periods were identified as the time between each sniff onset and subsequent peak. To quantify estimated perceived intermittency, the PID reading for each trial was sampled during inhalation periods. The perceived intermittency value was then calculated as described above with T = total inhalation time during the 6 s stimulus period.

## Optical imaging pre-processing

Imaging data was collected at 25 frames/s and 256×256 pixels, and pre-processed to correct for movement and global noise in every imaged frame. Glomerular ROIs were manually selected for each mouse accounting for glomeruli that may be recruited during different stimuli. Raw fluorescence traces were converted into ΔF/F using the average fluorescence of a 100 ms period prior to odor presentation as baseline fluorescence. Each trace was bandpass filtered (0.075–10 Hz, fourth-order Butterworth) to limit the contribution of noise to the measured response. Traces were baseline corrected for effects of photobleaching by fitting a second-degree polynomial to the response trace during pre- and post-odor periods. To obtain an estimate of the neural firing rate based on the GCaMP6f fluorescence calcium signal, filtered and baseline-corrected ΔF/F traces were deconvolved using a time constant of 150 ms (*Chen et al., 2013*).

## Identifying responding glomeruli

Estimated running firing rates (deconvolved ΔF/F traces) (FR) were z-scored relative to the baseline (2 s period preceding odor presentation) signal's mean and SD:

$$Z_{response} = \frac{FR - \mu_{pre-odor}}{\sigma_{pre-odor}}$$

For a given trial, if the glomerular response amplitude of the z-scored trace (identified as the trough to the peak of the response) associated with the first sniff of odor presentation exceeded a z-score value of 2, this glomerulus was identified as responding to odor for that specific trial. Sniff onsets were identified as previously described using the pressure sensor output and odor onsets were identified using the PID reading. Glomeruli that responded to >10% of trials (~6 trials) were included in the final quantification.

## Cross-correlation of glomerular response with odor dynamics

To calculate the glomerular response tracking of odor dynamics, deconvolved ΔF/F traces of each glomerulus were cross-correlated with the PID signal for a single trial. Correlation coefficients were calculated for lags between –500 ms and 500 ms using the xcorr() function in MATLAB. Both the deconvolved ΔF/F traces and PID signal are mean subtracted before calculating cross-correlations so that the reported coefficients represent the Pearson coefficient. For each glomerulus, we additionally calculated a shuffled average (10 shuffle iterations) in which the deconvolved ΔF/F traces were shuffled, cross-correlated with the PID reading, and correlation coefficients were averaged across the 10 shuffle iterations. This shuffled cross-correlation was subtracted from the original glomerular cross-correlation for each trial. A maximum correlation for each glomerulus was identified as the peak of the shuffle-subtracted cross-correlation for the given 500 ms window.

## Spatiotemporal analysis of response patterns

The z-scored response amplitude for each glomerulus was determined, as previously described, by measuring the difference in z-score value from trough to peak of the z-scored deconvolved ΔF/F glomerular response corresponding to the first sniff during odor presentation. Reported values are an average for each glomerulus over all trials. A T75 for each glomerulus was measured as the time to reach 75% of the response amplitude of the first sniff during odor presentation (as measured using % ΔF/F) from the sniff onset. Reported values are an average for each glomerulus over all trials.

Spatiotemporal response maps were established using the method outlined in *Baker et al., 2019*. Briefly, for a single trial, z-scored response amplitudes and T75 values of all glomeruli were correlated with their location along each of their spatial dimensions using corr() in MATLAB (A-P pixels and M-L pixels, with the latter being measured as the pixel distance from the midline). Reported values are an

average of the spatiotemporal response maps over all trials. The same approach was used to identify the olfactory bulb spatial location with the highest correlation between odor and glomerular response.

## Glomerular intermittency

GI was quantified using the z-scored deconvolved ΔF/F glomerular response traces. The threshold for intermittency quantification was set at a z-score value of 2 so that the GI was determined as the probability that the z-scored response trace is above 2 during the 6 s odor presentation period:

$$GI = Prob\left[Z_{response} \geq 2\right]$$

GI slope was quantified using a linear regression fit (fitlm() function in MATLAB) of the GI values vs odor intermittency across all trials for a given glomerulus. To determine if glomeruli significantly encoded intermittency as measured by GI representation across odor intermittency, a one-way ANOVA was performed using intermittency groups of 0.2, 0.5, and 0.8 for each glomerulus using all input trials (n=180 trials). A significant result was determined if the main effect of odor intermittency had p<0.001.

## Hierarchical clustering

To quantify similarities in glomerular responses across odor intermittency values, the average GI across odor intermittency values were correlated between all glomeruli. More specifically, each row in *Figure 5A*, *left,* was correlated with every other row to obtain the correlation coefficient values (r) represented in correlation matrix in *Figure 5B*, *middle*. If the GI of two glomeruli covary strongly in the same direction across odor intermittency values, the r value was close to 1. Hierarchical clustering was implemented using the Linkage() function in MATLAB by calculating correlation distance of the correlation matrix. Briefly, hierarchical clustering was determined using a method in which if 1 − r is close to 0, two glomeruli are quantified as closely related. Subsequent cluster analysis was performed using the cluster() function in MATLAB and a distance cutoff for cluster detection set (0.74).

## Linear classification

A threefold cross-validated linear discriminant classifier was implemented using the fitdiscr() function in MATLAB. The classifier was trained on one-third of all trials with equal sampling of each intermittency value (0.1–0.8) for each training set. The classifier was trained to predict the trial identity (CS+ or CS-) using GI as the predictors. The resulting classifier was then tested using the predict() function in MATLAB with remaining two-thirds of trials. Model accuracy and true positive rates were calculated as follows:

$$Accuracy = \frac{\#correct\ prediction}{\#test\ trials}$$

$$True\ Positive\ Rate = \frac{\#correct\ predicted\ CS+}{\#CS+trials}$$

Threefold cross-validation was repeated 20 times for each model to obtain error values for model prediction. For *Figure 5E*, responsive glomeruli were added at random for each iteration of 20 times threefold cross-validation. For anesthetized imaging experiments, the classifier was trained to predict the trial intermittency value (0.2, 0.5, and 0.8), as opposed to trial identity (CS+ or CS-).

## Acknowledgements

We appreciate the technical support of J Buckley and A Wilkins of the John B Pierce Shop. We thank Dr. Matt Wachowiak and Isaac Youngstrom for help with the sniff playback system. This project was supported by NIH/NIDCD grant R01 DC014723 and NSF BRAIN 1555880 to JV Verhagen and NSF BRAIN 1555862 to J Crimaldi. This project is supported by the NSF/CIHR/DFG/FRQ/UKRI-MRC Next Generation Networks for Neuroscience Program (Award #2014217). It was also supported by NIH NRSA 1F31DC018708 to A Gumaste.

## Additional information

### Funding

| Funder | Grant reference number | Author |
|---|---|---|
| National Science Foundation | NSF/CIHR/DFG/FRQ/UKRI-MRC Next Generation Networks for Neuroscience Program (Award #2014217) | Ankita Gumaste |
| National Institutes of Health | NRSA 1F31DC018708 | Ankita Gumaste |
| National Science Foundation | BRAIN 1555880 | Ankita Gumaste |
| National Science Foundation | BRAIN 1555862 | Aaron C True John P Crimaldi |
| National Institutes of Health | R01 DC014723 | Justus Verhagen |

The funders had no role in study design, data collection and interpretation, or the decision to submit the work for publication.

### Author contributions

Ankita Gumaste, Conceptualization, Data curation, Software, Formal analysis, Funding acquisition, Validation, Investigation, Visualization, Methodology, Writing – original draft, Project administration, Writing – review and editing; Keeley L Baker, Investigation, Writing – review and editing; Michelle Izydorczak, Resources, Investigation, Writing – review and editing; Aaron C True, Conceptualization, Resources, Writing – review and editing; Ganesh Vasan, Resources, Software, Methodology, Writing – review and editing; John P Crimaldi, Conceptualization, Resources, Supervision, Funding acquisition, Methodology, Writing – review and editing; Justus Verhagen, Conceptualization, Resources, Data curation, Software, Formal analysis, Supervision, Funding acquisition, Validation, Investigation, Visualization, Methodology, Writing – original draft, Project administration, Writing – review and editing

### Author ORCIDs

Aaron C True ⬥ http://orcid.org/0000-0001-9956-5105
Ganesh Vasan ⬥ http://orcid.org/0000-0002-6612-7739
Justus Verhagen ⬥ https://orcid.org/0000-0002-6090-0073

### Ethics

All procedures were performed in accordance with protocols approved by the Pierce Animal Care and Use Committee (PACUC) JV1-2019. These procedures are in agreement with the National Institutes of Health Guide for Care and Use of Laboratory Animals.

### Decision letter and Author response

Decision letter https://doi.org/10.7554/eLife.85303.sa1
Author response https://doi.org/10.7554/eLife.85303.sa2

## Additional files

### Supplementary files

• MDAR checklist

### Data availability

All data and analysis codes are available on Dryad.

The following dataset was generated:

| Author(s) | Year | Dataset title | Dataset URL | Database and Identifier |
|---|---|---|---|---|
| Verhagen J, Gumaste A | 2024 | Behavioral discrimination and olfactory bulb encoding of odor plume intermittency | https://doi.org/10.5061/dryad.crjdfn387 | Dryad Digital Repository, 10.5061/dryad.crjdfn387 |

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
