## [Editor Report]

This important work addresses the novel question for the vertebrate olfactory community of whether mice can discriminate odorant intermittency. The evidence supporting the conclusions is convincing. The authors used multiple experimental and analytical tools. The work will be of interest to sensory physiologists, both working in olfaction and navigation.

---

## [Decision Letter]

**Decision letter after peer review:**

Thank you for submitting your article "Behavioral discrimination and olfactory bulb encoding of odor plume intermittency" for consideration by *eLife*. Your article has been reviewed by 3 peer reviewers, one of whom is a member of our Board of Reviewing Editors, and the evaluation has been overseen by Piali Sengupta as the Senior Editor. The reviewers have opted to remain anonymous.

Essential revisions:

The reviewers agree that the hypothesis that animals can use intermittency on the odor plume as an important variable to help them discriminate olfactory stimuli is interesting and an important question to be answered. However, the experimental data does not fully support the author's claims.

1) The main concern about testing intermittency in the behavioral experiment is disambiguating it from concentration. The method used by the authors may not be sufficient to achieve this disambiguation. If the concentration is changed in a session-by-session manner, an animal may change its decision criteria and still perform the task based on the concentration, but not intermittence. Two ways of doing this: either to present different concentrations in the same session, i.e. scramble concentrations or to train animals on one concentration and present 'probe' trials with another concentration. If the behavior does not depend on concentration in any of these cases, it is strong evidence that the animal is using intermittency.

2) The way intermittency is measured is not sufficiently explained. The paper does not fully prove that the animal is sensitive to intermittency but has no other flow parameters. How the intermittency values are chosen is not clear. For example, low intermittency is considered to be below 0.15 and high intermittency is considered to be between 0.2 and 0.8. What is the range of intermittency values animals are exposed to in the wild? Is this known? Given the fact that intermittency is the main focus of this work, the authors plainly point out how intermittency is computed (in addition to the references to previous papers) and plainly write a sentence to give a quick grasp of what high and low intermittency means. If necessary a graphical description of the phenomenon would go a long way to more easily convey the information.

3) How to make sure that the stimulation with high intermittency values results in a higher odor concentration in the olfactory epithelium, resulting in ORNs responding to concentration, rather than other variables. The authors make a big effort to design a more naturalistic odorant stimulation, however, the selection of the two odorants used is not discussed. Are the odorants innately relevant for the animals? The degree of discrimination of intermittency changes when the odorant is appetitive. How about a neutral or aversive stimulus?

4) Work is needed on the explanation of exactly how the behavioral experiments are performed. For example, make it clear if they are using an odor A with low intermittency versus an odor B with high intermittency. What exactly are the odorants used for the CS+ and CS-, in addition to the different intermittency levels? Also, do the authors mix all stimuli in the same session, and so the animals simply generalize across all the stimuli and only consider intermittency for the behavioral choices? Or do authors repeat different sessions for different parameters?

5) General comments. There are several issues with how the figures are presented that if edited would help the reader to interpret the results much better. The methods section could be improved. A better description of the statistics used throughout the manuscript.

*Reviewer #1 (Recommendations for the authors):*

After carefully reading this work, I found it to be a very interesting work with relevant contributions to the field of sensory physiology. Nevertheless, I do have some comments that the authors need to address for this work to be fit for publication.

1) Although it is clear what the authors are referring to when using the term intermittency, I think that a graphical description of the phenomenon would go a long way to more easily convey the information. This could be incorporated into Figure 1.

2) Line 136. In this first experiment, animals are trained to low intermittency as the CS- and high intermittency as the CS+. I was wondering why not try both possibilities in the different animal cohorts. It would be possible that the "level" of intermittency affects how the animal interprets this parameter.

3) Although I celebrate the effort made to design more naturalistic odorant stimulation, is not explained the selection of the actual odorants used. Are the odorants innately relevant for the animals? The degree of discrimination of intermittency changes when the odorant is appetitive. How about a neutral or aversive stimulus?

4) Another question I have regarding the olfactory stimuli is the following. How about using a mixture of odorants? In the wild, animals are more likely to encounter stimuli comprised of multiple components. It would be very interesting to analyze this.

5) Line 230. The conclusion of these experiments is that the animals are capable of discriminating odors based on intermittency. I think the authors should work on making it clear if they are using an odor A with low intermittency versus an odor B with high intermittency. What exactly are the odorants used for the CS+ and CS-, in addition to the different intermittency levels?

6) Line 365. Please explain carefully the identity of the odorants used for the experiments that lead to the conclusion that glomeruli track odor stimuli. Please discuss this finding in more detail.

7) Some of the colors used in the figures were a bit hard to discriminate. If possible use a colorblind-friendly palette.

8) I would like to comment that I was very happily surprised to find in the methods sections that the authors have the animals in the correct light schedule. This makes me wonder if they took into account the time at which the experiments were conducted to evaluate if there is any kind of circadian regulation of intermittency discrimination. Maybe having a cohort of animals in constant darkness would be a good experiment if the circadian aspect of the question is of interest to the authors.

9) Line 273. Figure 2 – supp 1. Please discuss the increase in sniff frequency observed for the synthetic odors.

10) Line 480. Figure 5. I think that showing response maps of the different trials would go a long way to help the reader grasp the actual result.

11) Please elaborate more about the difference in the response of the ORN and mitral cells, and how these two different populations of cells can be encoders of intermittency with taking into account different properties.

12) The clustering of glomeruli in 2 clusters needs clarification. It would be possible to show a map of the different clusters in multiple animals?

*Reviewer #2 (Recommendations for the authors):*

1) I kept having doubts throughout the paper about how intermittency was computed and what high or low intermittency meant. Given that intermittency is the focus of the whole paper, I would suggest the authors plainly point out how intermittency is computed (in addition to the references to previous papers) and plainly write a sentence to give a quick grasp of what high and low intermittency means.

2) Figures are not well organized in terms of both content and graphic.

Content-wise: I found it very unintuitive and challenging to find the specific subfigure supporting specific claims. I had to juggle back and forth between the main and supplementary figures continuously. Some key results are presented in supplementary figures instead of main figures. The figures should be organized in chronological citation order, and key results/claims should be supported by at least one main figure item. All figures and subplots should be referenced in the text.

Graphic-wise: the font character is very small and hard to read, and some color-legend and axis labels are missing.

3) Please improve Figure 2A so that we can well understand the computation of the estimated perceived intermittency. For example, use the same temporal scale for the original signal and the estimated perceived intensity so that we can directly compare.

4) Figure 4D: please plot the average behavioral accuracy so that we can visualize the claim made in the text.

5) Whenever you report p-values, please, specify the statistical test you have used.

*Reviewer #3 (Recommendations for the authors):*

The manuscript could be further improved by addressing the following points.

1. The authors conclude from the behavioral data in Figure 1 that mice can discriminate between fluctuating odor stimuli based on intermittency values. However, a high intermittency value also means a higher total amount of odorants delivered (sniffed). Even though the concentration of the delivered odor is kept constant, depending on the properties of the odor (e.g., its absorption in the mucous), the actual concentration OSNs encounter may be different with varying intermittency values. This raises the possibility that the mice may discriminate other parameters rather than intermittency values per se. The authors should at least discuss alternative interpretations of these results.

2. Related to #1, the choice of the two odors is not explained or well justified. The chemical and physical properties of an odor may influence the final concentration the OSNs encounter and the behavioral outcome. Testing a few more odorants with different properties would be informative and help to strengthen the conclusions related to the odor-specific encoding of intermittency (Line 588) and spatial patterning/intrinsic glomerular properties (Line 358-360).

3. Line 527-530. "OSNs show a 13.4% decrease in the number of ….from 2-8 Hz sniff frequency (Figure 5A, one-way ANOVA)". Figure 5A does not show this decrease of 13.4%. If this were shown in Figure 5B, the difference in the purple bars between 2 and 8 Hz is more like >40%. For the M/T cells, a 25.2% increase is stated in the main text, but the green bar's increase from 2 to 8 Hz is more like 30%.

4. Line 530-532. When describing 2-heptanone results, please cite Figure 6A, B. Again double check the values of 22.4% and 25.1% increase from 2 to 8 Hz sniff frequency for OSNs and M/T cells. From Figure 6B, for OSNs (purple), there seems a slight decrease from 2 to 8 Hz.

[Editors' note: further revisions were suggested prior to acceptance, as described below.]

Thank you for resubmitting your work entitled "Behavioral discrimination and olfactory bulb encoding of odor plume intermittency" for further consideration by *eLife*. Your revised article has been evaluated by Piali Sengupta (Senior Editor), a Reviewing Editor, and the original reviewers.

The manuscript has been improved but there are some remaining issues that need to be addressed, as outlined below:*Reviewer #2:*

I had a chance to look carefully into behavioral part of the paper, and I found the following problems:

1) The authors are trying to make a case that the animal is attending to the intermittence variable. It is a challenging problem, and I do not think that the authors succeeded.

The Figure 2B, shows different performances for high and low gain stimuli. The shift between two curves may be explained by an alternative animal strategy. Let's assume that an animal is doing a concentration discrimination task. It estimates an integral of concentration and respond as 'go', if its value is above some threshold. I can imagine that the behavioral results for such a task for stimuli with high and low gain presented in coordinates of intermittency would look like as shown at Figure 2B. For the low gain, an animal reach maximum perforce at a higher level of intermittency, exactly as shown on the figure.

I would actually strongly suggest that authors would plot their results as a function, of integral of concentration.

2) The authors tried to provide an argument for an intermittency by presenting a predicted performance based on concentration Figure S2B. But I could not find explanations of how this is estimated.

3) In general, an animal is computing some combination of variables, and it is extremely difficult to point out which one is most relevant. So far, the focus on intermittency is not motivated.

4) In the previous review I asked to provide the logic for choosing a specific value for intermittency threshold. I did not fully follow their explanation. The authors presented the graph of intermittency as a function of distance (Figure S1A). If the logic that an animal is using an intermittency as a measure of a distance to the source, then the measurement should be done at the interval of intermittencies where it is mostly informative about the distance. Based on the figure, this should be around 0.5, where the slope of intermittency as a function of distance is the steepest. It would be much more reasonable to show an animal's ability to differentiate intermittency around this level. The ability to discriminate intermittency at the level of 0.15 tells us only that an animal can only extract information that the source is far away. This can be potentially done using other cues.

---

## [Author Response]

Essential revisions:The reviewers agree that the hypothesis that animals can use intermittency on the odor plume as an important variable to help them discriminate olfactory stimuli is interesting and an important question to be answered. However, the experimental data does not fully support the author's claims.1) The main concern about testing intermittency in the behavioral experiment is disambiguating it from concentration. The method used by the authors may not be sufficient to achieve this disambiguation. If the concentration is changed in a session-by-session manner, an animal may change its decision criteria and still perform the task based on the concentration, but not intermittence. Two ways of doing this: either to present different concentrations in the same session, i.e. scramble concentrations or to train animals on one concentration and present 'probe' trials with another concentration. If the behavior does not depend on concentration in any of these cases, it is strong evidence that the animal is using intermittency.

We appreciate the reviewer pointing out our oversight in including this information in the manuscript. Trials of the two gain values (which modulate the maximum concentration) are presented interleaved within a session. These trials are solely separated for post-session analysis to test the effect of gain on animal performance. To make this point clearer we have included the following text in the “Go/No-Go behavior” subsection of the Methods section of the manuscript on line 952-955:

“Additionally, trials of a gain of 0.5 and a gain of 1 are interwoven randomly during the session with each unique stimulus being presented at both a gain of 0.5 and 0.1. Thus, after the initial engagement trials, animals are presented with a total of 28 trials at a gain of 0.5 and 28 trials at a gain of 0.1.”2) The way intermittency is measured is not sufficiently explained. The paper does not fully prove that the animal is sensitive to intermittency but has no other flow parameters. How the intermittency values are chosen is not clear. For example, low intermittency is considered to be below 0.15 and high intermittency is considered to be between 0.2 and 0.8. What is the range of intermittency values animals are exposed to in the wild? Is this known? Given the fact that intermittency is the main focus of this work, the authors plainly point out how intermittency is computed (in addition to the references to previous papers) and plainly write a sentence to give a quick grasp of what high and low intermittency means. If necessary a graphical description of the phenomenon would go a long way to more easily convey the information.

We have broken up our response to this comment based on the specific questions asked:

1. The way intermittency is measured is not sufficiently explained. Given the fact that intermittency is the main focus of this work, the authors plainly point out how intermittency is computed (in addition to the references to previous papers) and plainly write a sentence to give a quick grasp of what high and low intermittency means. If necessary a graphical description of the phenomenon would go a long way to more easily convey the information.

In order to clarify how intermittency is computed, we made a new insert and legend to Figure 1. We addressed this with Reviewer 1 issue 1 and Reviewer 2 issue 1. We are open to suggestions if this schematic needs further clarification.

2. The paper does not fully prove that the animal is sensitive to intermittency but has no other flow parameters.

We do not believe the reviewers raise the issue of flow, but nonetheless our olfactometer was designed to minimize changes in flow in context of dynamically varying odor concentration by its symmetric push-pull constant flow design (Figure 1B, Methods lines 762-768). Thus, no parameters other than fluctuations in odor concentration are provided to the animal. We have stated the following in our methods section:

“Additionally, the carbon-filtered nitrogen line was split before passing through the odor and is directed to an empty PFA vial used for counterbalancing odor flow. Both odor delivery and counterbalanced nitrogen delivery are controlled by independent EVP Series Clippard Proportional Valves and an EVPD-2 valve driver (Clippard Instrument Laboratory, Inc, Cincinnati, Ohio). Proportional valves were calibrated so as to ensure that final combined nitrogen and odor flow is maintained at 50 mL/min. Combined nitrogen and odor flow was confirmed using a flow meter (Omron Electronic Components Product # D6F-P0010A1).”

Further we addressed under Reviewer 3 issue 1a that odor molecule absorption temporal integration can explain only some of the discriminatory psychometric curves. Thus, we can't think of or speculate on other potentially relevant parameters co-varying with intermittency other than mechanistic downstream processes inside the mouse itself to enable such discrimination.

3. How the intermittency values are chosen is not clear. For example, low intermittency is considered to be below 0.15 and high intermittency is considered to be between 0.2 and 0.8. What is the range of intermittency values animals are exposed to in the wild? Is this known?

We addressed this comment under Reviewer 2 issue 2 and 3, including with a new Figure 1 Supplementary Figure 1A.

3) How to make sure that the stimulation with high intermittency values results in a higher odor concentration in the olfactory epithelium, resulting in ORNs responding to concentration, rather than other variables. The authors make a big effort to design a more naturalistic odorant stimulation, however, the selection of the two odorants used is not discussed. Are the odorants innately relevant for the animals? The degree of discrimination of intermittency changes when the odorant is appetitive. How about a neutral or aversive stimulus?

We have broken up our response to this comment based on the specific questions asked:

1. How to make sure that the stimulation with high intermittency values results in a higher odor concentration in the olfactory epithelium, resulting in ORNs responding to concentration, rather than other variables.

We address this under Reviewer 3 issue 1. Indeed, part of the mechanism appears to be integration over time, but not all by far. This is also illustrated in Figure 2- Supplement 1.

2. The authors make a big effort to design a more naturalistic odorant stimulation, however, the selection of the two odorants used is not discussed. Are the odorants innately relevant for the animals? The degree of discrimination of intermittency changes when the odorant is appetitive. How about a neutral or aversive stimulus?

We addressed this under Reviewer 1 issue 3 and 4, Reviewer 2 Public issue 4, Reviewer 3 issue 2. The study was designed to explore a plethora of intermittency-related parameters rather than odorant-related issues. We selected two fruit-associated odors with neutral preference indices with a large comparative literature behind them, while using 3 odor trace types at 2 gains in awake and anesthetized mice expressing GcamP6f under OMP or Thy1 promoter. We also explored varying sniff rates. We agree further research should explore a wider array of odorants, including mixtures.

4) Work is needed on the explanation of exactly how the behavioral experiments are performed. For example, make it clear if they are using an odor A with low intermittency versus an odor B with high intermittency. What exactly are the odorants used for the CS+ and CS-, in addition to the different intermittency levels? Also, do the authors mix all stimuli in the same session, and so the animals simply generalize across all the stimuli and only consider intermittency for the behavioral choices? Or do authors repeat different sessions for different parameters?

We agree that additional explanation was needed on the order and specific conditions of the behavior experiments. To answer the specific questions asked by reviewers:

1. What exactly are the odorants used for the CS+ and CS-, in addition to the different intermittency levels?

We further clarifed on line 965-970: "All animals are trained and tested using 2% methyl valerate in mineral oil (Σ Aldrich product #1489977). After completing the entire behavioral paradigm (testing using naturalistic, binary naturalistic, and square-wave stimulus sets) using 2% methyl valerate, animals are tested on the binary naturalistic condition using 2% 2-heptanone in mineral oil (Σ Aldrich product #537683). Thus, within a session, all CS+ and CS- are a single odor, the distinguishing property between CS+ and CS- are their intermittency value."

2. Also, do the authors mix all stimuli in the same session, and so the animals simply generalize across all the stimuli and only consider intermittency for the behavioral choices? Or do authors repeat different sessions for different parameters?

Stimuli presented within each session all come from one stimulus set type (e.g. all 64 trials in a session are either naturalistic, binary naturalistic, or square-wave stimuli). As stated in the response to Essential Revision #1, stimulus gain is mixed within session. Animals were tested using two unique sets of naturalistic, binary naturalistic, and square-wave stimuli each. We further added line 959-963: "The order in which animals were tested on each stimulus set type was randomly permuted (e.g. some animals were first tested on naturalistic and once criteria was met, they were moved onto square wave, and then lastly onto binary naturalistic. Other animals were started on binary naturalistic and once criteria was met, they were moved onto square-wave, and then lastly onto naturalistic).

We have clarified the behavioral experiment procedure by breaking up the “Go/No-Go behavior” subsection of the methods section into several parts following the order in which the behavioral training and testing was performed. We have included the above information in the “Complete Go/No-Go task” heading of the “Go/No-Go behavior” subsection.

5) General comments. There are several issues with how the figures are presented that if edited would help the reader to interpret the results much better. The methods section could be improved. A better description of the statistics used throughout the manuscript.

We have edited the figures according to comments brought up in subsequent sections. We have reorganized figures where appropriate, addressed accessibility issues, added clarifying figures, and increased font size where possible.

In general we have expanded the methods sections at various points. The most notable addition to the methods section includes an in-depth description of the Go/No-Go behavioral task. In addition, we have addressed the description of statistics below. We however, may need more specifics to fully address the comment “the methods section could be improved,” if our described changes do not meet expectations.

We have included detailed statistical information in-text (including all regression model equations, generalized linear model equations, etc.) with additional information in figure legends. For all statistical tests we have stated the test run, the relevant statistical outputs (model equations, r^2^ values, etc.), and the p-value. We are happy to provide more information on these statistics but require more specification from the review on what constitutes a “better description.”

We have added additional information (lines 1014-1016 and 1067-1125) in the “Quantification and Statistical Analyses” section on the MATLAB functions used to implement each statistical test used in the manuscript.

Reviewer #1 (Recommendations for the authors):After carefully reading this work, I found it to be a very interesting work with relevant contributions to the field of sensory physiology. Nevertheless, I do have some comments that the authors need to address for this work to be fit for publication.1) Although it is clear what the authors are referring to when using the term intermittency, I think that a graphical description of the phenomenon would go a long way to more easily convey the information. This could be incorporated into Figure 1.

We have edited the figures according to comments brought up in subsequent sections. We have reorganized figures where appropriate, addressed accessibility issues, added clarifying figures, and increased font size where possible.

In general we have expanded the methods sections at various points. The most notable addition to the methods section includes an in-depth description of the Go/No-Go behavioral task. In addition, we have addressed the description of statistics below. We however, may need more specifics to fully address the comment “the methods section could be improved,” if our described changes do not meet expectations.

We have included detailed statistical information in-text (including all regression model equations, generalized linear model equations, etc.) with additional information in figure legends. For all statistical tests we have stated the test run, the relevant statistical outputs (model equations, r^2^ values, etc.), and the p-value. We are happy to provide more information on these statistics but require more specification from the review on what constitutes a “better description.”

We have added additional information (lines 1014-1016 and 1067-1125) in the “Quantification and Statistical Analyses” section on the MATLAB functions used to implement each statistical test used in the manuscript.

2) Line 136. In this first experiment, animals are trained to low intermittency as the CS- and high intermittency as the CS+. I was wondering why not try both possibilities in the different animal cohorts. It would be possible that the "level" of intermittency affects how the animal interprets this parameter.

We appreciate this feedback for a future direction for our lab, which would highlight the robust nature of intermittency discrimination in mice. Although not included in this manuscript, we did perform a pilot experiment including the reverse paradigm (high intermittency as CS- and low intermittency as CS+). While we will not include these preliminary results in the present manuscript, we did find that the animals were able to perform the task with the paradigm reversed. In order to streamline behavioral testing and result interpretation, we chose to pursue one form of the paradigm. We chose to test low intermittency as CS- and high intermittency as CS+ as being rewarded at high intermittency values may parallel animals’ behavior when their goal is to navigate towards an odor source. However, we do acknowledge that testing the reverse paradigm would further support our finding that mice can discriminate odor intermittency.

3) Although I celebrate the effort made to design more naturalistic odorant stimulation, is not explained the selection of the actual odorants used. Are the odorants innately relevant for the animals? The degree of discrimination of intermittency changes when the odorant is appetitive. How about a neutral or aversive stimulus?

These odorants (methyl valerate and 2-heptanone) were selected in part due to their spatial representation in the olfactory bulb. Both methyl valerate and 2-heptanone activate glomeruli on the dorsal surface of the olfactory bulb, therefore allowing us to image neural activity in response to presentation of these odorants (as the optical window procedure is performed on the dorsal olfactory bulb). Additionally, these two odorants recruit glomeruli in different regions of the dorsal olfactory bulb, have different functional groups and elicit different spatiotemporal response properties in the olfactory bulb (Figure 6—figure supplement 1A, stated on line 507).

Both odorants are fruit-associated odors with neutral preference indices (Saraiva et al., 2016, Fletcher, 2012). Thus, while we do not explore a panel of odorants, we do explore the generalizability of intermittency processing with two distinct odorants.

We have included more information on the odorant selection in the “Sniff frequency-dependent glomerular representation of intermittency” section of the Results section on line 505-507.

4) Another question I have regarding the olfactory stimuli is the following. How about using a mixture of odorants? In the wild, animals are more likely to encounter stimuli comprised of multiple components. It would be very interesting to analyze this.

We appreciate the suggestion and indeed we considered a mixture for the reasons suggested, however given the many variables we tested in this experiment did not have the capacity to introduce odor mixtures. We are, unfortunately, not aware of a "standard" odor mixture (e.g. from fragrance/food industry) that has been widely researched in rodents.

5) Line 230. The conclusion of these experiments is that the animals are capable of discriminating odors based on intermittency. I think the authors should work on making it clear if they are using an odor A with low intermittency versus an odor B with high intermittency. What exactly are the odorants used for the CS+ and CS-, in addition to the different intermittency levels?

We appreciate the reviewer pointing out this area for further clarification. We have added additional clarification on the behavioral paradigm in the “Go/No-Go behavior” subsection of the Methods section. To answer this specific question we added on Line 965-970: "All animals are trained and tested using 2% methyl valerate in mineral oil (Σ Aldrich product #1489977). After completing the entire behavioral paradigm (testing using naturalistic, binary naturalistic, and square-wave stimulus sets) using 2% methyl valerate, animals are tested on the binary naturalistic condition using 2% 2-heptanone in mineral oil (Σ Aldrich product #537683). Thus, within a session, all CS+ and CS- are a single odor, the distinguishing property between CS+ and CS- are their intermittency value.".

6) Line 365. Please explain carefully the identity of the odorants used for the experiments that lead to the conclusion that glomeruli track odor stimuli. Please discuss this finding in more detail.

As stated in the “Spatial Mapping of Glomerular Response Properties” subsection of the Results section,we use methyl valerate for these experiments. We have now stated this information earlier in this subsection in line 363 and 366. Additionally, we have added this information under the “Awake Wide-Field Calcium Imaging” header of the “Go/No-Go Behavior” subsection of the Methods section, line 972-978:

"Awake Wide-Field Calcium Imaging

After being tested on the 2-heptanone control condition, animals underwent an optical window procedure and were allowed a minimum 48-hour recovery period. After the recovery period, animals were tested on the binary naturalistic and square-wave stimulus sets using methyl valerate. The order in which animals were tested on each stimulus set type was randomly permuted. During this testing period, the dorsal olfactory bulbs of these animals were imaged according to the optical imaging protocol described in the “Optical Imaging System” section." In using a single odor for optical imaging, we prioritized being able to image animals on multiple stimulus sets (binary naturalistic and square-wave) given the limited window of time before a decline in the optical window clarity.

7) Some of the colors used in the figures were a bit hard to discriminate. If possible use a colorblind-friendly palette.

All of our figures have now been tested with online colorblind-checkers to ensure that colors are distinguishable. We have changed colors appropriately to accommodate these accessibility needs.

8) I would like to comment that I was very happily surprised to find in the methods sections that the authors have the animals in the correct light schedule. This makes me wonder if they took into account the time at which the experiments were conducted to evaluate if there is any kind of circadian regulation of intermittency discrimination. Maybe having a cohort of animals in constant darkness would be a good experiment if the circadian aspect of the question is of interest to the authors.

We appreciate the idea, but we unfortunately do not have additional insights into this or data to evaluate this further.

9) Line 273. Figure 2 – supp 1. Please discuss the increase in sniff frequency observed for the synthetic odors.

Firstly, we appreciate the reviewer drawing attention to this as this figure caption was mislabeled and should read “Square-wave” instead of “Synthetic.” We have made the change in the manuscript. The increase in sniff frequency for miss trials at an intermittency value on 0.5 for the square-wave stimulus set should be interpreted with caution as there was only one session with miss trials for this intermittency value, therefore this single point is not representative of an average response. We do not attribute any specific interpretation to this increase as overall, the graphs indicate that when multiple sessions are averaged, there is no effect of trial outcome or intermittency on average trial sniff frequency.

10) Line 480. Figure 5. I think that showing response maps of the different trials would go a long way to help the reader grasp the actual result.

We have now included example response maps of t75 and z-score amplitude for both methyl valerate and heptanone presentation for OMP-GCaMP6f and THY1-GCaMP6f animals in Figure 6—figure supplement 1A *right*.

11) Please elaborate more about the difference in the response of the ORN and mitral cells, and how these two different populations of cells can be encoders of intermittency with taking into account different properties.

We appreciate the interesting and relevant issue raised, but still think it would be too speculative to relate the complex and diverse ORN and MTC response properties and OB transformations, as currently understood, to intermittency. These are comprehensively described in the following reference: Martelli C, Storace DA. (2021). Stimulus driven functional transformations in the early olfactory system. Frontiers in Cellular Neuroscience. Aug 3; DOI: 10.3389/fncel.2021.684742.

12) The clustering of glomeruli in 2 clusters needs clarification. It would be possible to show a map of the different clusters in multiple animals?

We have added three examples to Figure 5- Supplement 1 (the former Figure 4- Supplement 1 has been moved to Figure 5- Supplement 2).

Reviewer #2 (Recommendations for the authors):1) I kept having doubts throughout the paper about how intermittency was computed and what high or low intermittency meant. Given that intermittency is the focus of the whole paper, I would suggest the authors plainly point out how intermittency is computed (in addition to the references to previous papers) and plainly write a sentence to give a quick grasp of what high and low intermittency means.

We agree with you and reviewer 1 on this – thank you. We hence made this clarifying schematic and legend.

2) Figures are not well organized in terms of both content and graphic.Content-wise: I found it very unintuitive and challenging to find the specific subfigure supporting specific claims. I had to juggle back and forth between the main and supplementary figures continuously. Some key results are presented in supplementary figures instead of main figures. The figures should be organized in chronological citation order, and key results/claims should be supported by at least one main figure item. All figures and subplots should be referenced in the text.Graphic-wise: the font character is very small and hard to read, and some color-legend and axis labels are missing.

We have now reorganized many of our figures and created new figures. In addition, we have increased the font size on all figures where possible. We hope that these changes have increased the readability of the manuscript.

3) Please improve Figure 2A so that we can well understand the computation of the estimated perceived intermittency. For example, use the same temporal scale for the original signal and the estimated perceived intensity so that we can directly compare.

The temporal scale in Figure 3A (new figure label) is consistent between the original signal and the estimated perceived odor. Thus, these two traces can directly be compared. The figure is illustrating that the estimated perceived odor is extracted from the inhalation periods of the raw trace (assuming that animals are not perceiving odor that they are not inhaling). We intentionally left off the scale in this figure to not mislead the reader into thinking that somehow the estimated perceived odor is a faster signal (when in fact it is the same odor signal as the original, simply just extracted during inhalation periods). Thus, the perceived graph is exactly the original graph but with exhalation periods removed, as retention of original inhalation timing is not relevant to the analyses.

4) Figure 4D: please plot the average behavioral accuracy so that we can visualize the claim made in the text.

We believe the reviewer may be referring to Figure 5E (former Figure 4E, where we note “Just 22 glomeruli with high GI slopes are enough to predict trial outcome at the same accuracy as the average animal hit rate”). The average hit rate was previously represented on the graph as a dotted line (indicated as such in the figure legend). Please inform us if there is additional information missing.

5) Whenever you report p-values, please, specify the statistical test you have used.

We have ensured that all specifics associated with statistical tests are reported in-text within the Results section along with p-values. The p-values are also stated in the figure legends (consistent with our previous draft), and we have refrained from editing the legends to also reflect this statistical test information in order to avoid lengthy figure legends including extensive model notation.

Reviewer #3 (Recommendations for the authors):The manuscript could be further improved by addressing the following points.1. The authors conclude from the behavioral data in Figure 1 that mice can discriminate between fluctuating odor stimuli based on intermittency values. However, a high intermittency value also means a higher total amount of odorants delivered (sniffed). Even though the concentration of the delivered odor is kept constant, depending on the properties of the odor (e.g., its absorption in the mucous), the actual concentration OSNs encounter may be different with varying intermittency values. This raises the possibility that the mice may discriminate other parameters rather than intermittency values per se. The authors should at least discuss alternative interpretations of these results.

This is a really interesting point made by the reviewer and we thank you for it. To be sure the concentration was varied over time in several ways: 1. During a trial: Continuously vs Binary 2. Across trials: high vs low gain (Figure 1F). We were indeed curious whether mice could discriminate merely based on the amount of "deposited molecules" as one would expect intermittency to affect. To test this "integration" hypotheses we halved the amount of odorant molecules delivered. We found that while this did indeed shift the psychometric curve in an expected direction, however it fell short of the shift one would predict if integration were the sole mechanism (Figure 2—figure supplement 1B). Line 233-240:

"As mentioned, mice were tested on two gain values to determine the degree to which odor concentration integration affected their task decisions. Although there was an effect of gain on behavioral performance, animal performance at 0.5 gain was significantly better than a psychometric curve prediction of animal performance solely based on odor concentration integration (Figure 2—figure supplement 1B, Mixed effects model, n = 48 sessions: Performance ~ Intermittency + Genotype + Stimulus Type + Gain; Main effect of Gain, p=0.00013, one-tailed ttest with Bonferroni correction, Naturalistic intermittency ≥ 0.3, p<0.0001; Naturalistic intermittency ≥ 0.5, p<0.0001; Square-wave intermittency ≤ 0.8, p<0.0001)."

Our olfactory bulb imaging aims to further address what neural correlates exist to the behavioral performance, where the answer appears to be glomerulus-dependent (and to some extent odor-dependent).

2. Related to #1, the choice of the two odors is not explained or well justified. The chemical and physical properties of an odor may influence the final concentration the OSNs encounter and the behavioral outcome. Testing a few more odorants with different properties would be informative and help to strengthen the conclusions related to the odor-specific encoding of intermittency (Line 588) and spatial patterning/intrinsic glomerular properties (Line 358-360).

We much appreciate the concerns of the reviewer.

We indeed considered several odorants and associated properties. Given time constrains we were limited to 2 stimuli of which we had to vary many parameters (type, I, gain, sniffing) in assessing both discrimination and neural processing.

We decided to test 2 monomolecular odorants (2-heptanone and methyl valerate) as these have been most widely used in rodent olfactory bulb imaging, providing distinct and clear glomerular response patterns. They are both fruity smelling odors, implying a relationship to edible food (at least, for humans). Methyl valerate is a methyl ester of pentatonic acid with a fruity (apple) smell. 2-Heptanone is a ketone with a fruity (green banana) smell.

Nonetheless we agree exploring additional odorants with very different perceptual or physico-chemical properties will be of great interest, as this could affect the psychometric functions, associated active sampling behavior and bulbar response patterns. However, given the extensive testing on many parameters, we did not have the capacity to introduce more odors. Nonetheless, we are confident that mice in principle can discriminate intermittency.

3. Line 527-530. "OSNs show a 13.4% decrease in the number of ….from 2-8 Hz sniff frequency (Figure 5A, one-way ANOVA)". Figure 5A does not show this decrease of 13.4%. If this were shown in Figure 5B, the difference in the purple bars between 2 and 8 Hz is more like >40%. For the M/T cells, a 25.2% increase is stated in the main text, but the green bar's increase from 2 to 8 Hz is more like 30%.

We appreciate the reviewer noting that we had incorrectly referenced Figure 5A, when indeed, we intended to reference Figure 5B (we have made this correction in the manuscript). However, the change values previously reported are accurate. We have now included the raw values as well on lines 557-561.

4. Line 530-532. When describing 2-heptanone results, please cite Figure 6A, B. Again double check the values of 22.4% and 25.1% increase from 2 to 8 Hz sniff frequency for OSNs and M/T cells. From Figure 6B, for OSNs (purple), there seems a slight decrease from 2 to 8 Hz.

We have now referenced figures 6A and B in this section. We have confirmed that the reported values for the change in intermittency-encoding glomeruli from 2 to 8 Hz for both OSNs and M/T cells were accurate. As we did in the above case, we have now reported the raw values as well on line 564.

[Editors’ note: what follows is the authors’ response to the second round of review.]

The manuscript has been improved but there are some remaining issues that need to be addressed, as outlined below:Reviewer #2:I had a chance to look carefully into behavioral part of the paper, and I found the following problems:1) The authors are trying to make a case that the animal is attending to the intermittence variable. It is a challenging problem, and I do not think that the authors succeeded.The Figure 2B, shows different performances for high and low gain stimuli. The shift between two curves may be explained by an alternative animal strategy. Let's assume that an animal is doing a concentration discrimination task. It estimates an integral of concentration and respond as 'go', if its value is above some threshold. I can imagine that the behavioral results for such a task for stimuli with high and low gain presented in coordinates of intermittency would look like as shown at Figure 2B. For the low gain, an animal reach maximum perforce at a higher level of intermittency, exactly as shown on the figure.I would actually strongly suggest that authors would plot their results as a function, of integral of concentration.

The reviewer is making a valid point – can the discrimination be explained by another

(unintended) correlated variable. We anticipated this issue, as with increased intermittency, i. e. higher duty cycle, mice are exposed to the same concentration for a larger fraction of time. This could, and likely does to some degree, increase the absorbed amount of odor and hence increase perceived odor intensity. This forms the basis for the reviewer’s "integral of concentration", which stands as an alternative to time-cue related mechanisms (odor onset and offset times, duration of inhalations when no odor is present, etc) solely reflecting when the stimulus is suprathreshold with no concern for intensity.

Indeed, to explore to what extent mice relied on such integration we designed the task with 2 levels of odor concentration (L 183-186; Figure 1F, Figure 2B): the lower concentration (gain 0.5) being roughly half of the higher (gain 1; Figure S 1D). " To test the degree to which odor concentration integration may inform decisions on the intermittency discrimination task, mice were tested on interleaved trials using a two gain values, where in trials with a gain of 0.5, the maximum stimulus concentration was halved (Figure 1F, Figure 1—figure supplement 1D)."

We reasoned that if mice are solely relying on intensity-integration, then halving the odor concentration (gain 0.5) would be fully equivalent to halving the intermittency at gain 1, as the amount of total absorbed odorant during a trial would be identical in both cases. Hence the expected psychometric curves would be right-shifted so that performance at gain 1 I=0.2 is moved to I=0.4, 0.3 to 0.6, 0.4 to 0.8. These straightforward hypothetical concentration integration curves are the black lines shown on Figure S 2B.

Our results (L 239-242) show that these two cases are not equivalent behaviorally: "Although there was an effect of gain on behavioral performance, animal performance at 0.5 gain was significantly better than a psychometric curve prediction of animal performance solely based on odor concentration integration". Figure 2B shows qualitatively that the psychometric curves at gain 0.5 are not simply 2x right-shifted versions of gain=1. The caption further states " At gain 1, mice perform significantly above chance at intermittency values of 0.3 and above (one-tailed ttest, Bonferroni correction, p<0.0001, n=48 sessions) for all stimulus types. At gain 0.5, mice perform above chance at intermittency values 0.4 and above, naturalistic and square-wave, 0.5 and above, binary naturalistic (one-tailed t-test, Bonferroni correction, p<0.0001, n=48 sessions)." Thus, statistically the intermittency discrimination thresholds shift much less (0.4, 0.4 and 0.5) than the expected doubling of 0.3 (i.e., 0.6). This further demonstrates that concentration-integration can explain only some the behavior.

Based on the reviewers comments we have now further clarified in the text how the psychometric curves at both gains have provided evidence that concentration integration plays some, but only some, role in discrimination dynamic odor stimuli with varying physical intermittencies. We added LL232-239: " If mice are solely relying on intensity-integration, then halving the odor concentration (gain=0.5) would be fully equivalent to halving the intermittency at gain=1, as the amount of total absorbed odorant during a trial would be identical in both cases. Figure 2B demonstrates that psychometric curves for gain=0.5 are not right-shifted versions of gain=1 by the expected equivalent doubling of intermittency. Indeed, the intermittency discrimination thresholds shifted much less (0.4, 0.4 and 0.5) than the expected doubling of 0.3 (i.e., 0.6)."

In subsequent text we statistically show that the behavioral curves do not match the predicted concentration-integration curves, as described in our prior round to reviewer 3. Indeed, Reviewer 3 made a very similar remark in the first round of reviewing as their Issue 1, to which our response has been accepted as satisfactory and we copy below:

"This is a really interesting point made by the reviewer and we thank you for it. To be sure the concentration was varied over time in several ways: 1. During a trial: Continuously vs Binary 2. Across trials: high vs low gain (Figure 1F). We were indeed curious whether mice could discriminate merely based on the amount of "deposited molecules" as one would expect intermittency to affect. To test this "integration" hypotheses we halved the amount of odorant molecules delivered. We found that while this did indeed shift the psychometric curve in an expected direction, however it fell short of the shift one would predict if integration were the sole mechanism (Figure 2—figure supplement 1B). Line 233-240:" As mentioned, mice were tested on two gain values to determine the degree to which odor concentration integration affected their task decisions. Although there was an effect of gain on behavioral performance, animal performance at 0.5 gain was significantly better than a psychometric curve prediction of animal performance solely based on odor concentration integration (Figure 2—figure supplement 1B, Mixed effects model, n = 48 sessions: Performance ~ Intermittency + Genotype + Stimulus Type + Gain; Main effect of Gain, p=0.00013, one-tailed t-test with Bonferroni correction, Naturalistic intermittency ≥ 0.3, p<0.0001; Naturalistic intermittency ≥ 0.5, p<0.0001; Square-wave intermittency ≤ 0.8, p<0.0001)."

Our olfactory bulb imaging aims to further address what neural correlates exist to the behavioral performance, where the answer appears to be glomerulus-dependent (and to some extent odor-dependent)."

We also like to point the reviewer to the **Discussion** of the findings pertaining to gain (L655675), where we fully acknowledge some role for the reviewers alternative explanation: " We found that mouse performance on the intermittency discrimination task is not affected by the odor used or frequency of odor whiffs, but is affected by the concentration gain. This shows that intermittency is a temporal property of odor plumes that can be detected independently from other temporal properties, such as whiff frequency. This distinction may be important if different temporal properties provide at least partially independent information about location within the odor plume, as suggested by Jayaram et al. (2022). Other temporal properties that may indicate distance from and composition of an odor source are odor whiff frequency and the temporal correlation of fluctuating odors, both of which mice are capable of detecting (Hopfield, 1991; Schmuker et al., 2016; Ackels et al., 2021; Dasgupta et al., 2022). It is possible that these temporal properties are either used independently or in concert during odor navigation. Additionally, we show that concentration gain has an effect on intermittency discrimination, suggesting that mice are in part using odor concentration integration for intermittency discrimination. While further work needs to be done to explore discrimination of odor stimuli based on odor integration, a plethora of work suggests that rodents can discriminate odor duration and intensity both at the neural and behavioral levels (Rubin et al., 1999; Rospars et al., 2000; Spors et al., 2002; Li et al., 2014; Wojcik et al., 2014; Sirotin et al., 2015; Li et al., 2020). In some odor plumes, odor concentration and odor intermittency both increase as distance from the odor source decreases, indicating that the integral of measured odor would also increase (Connor et al., 2018). It is possible that odor intermittency and odor integration might inform odor source localization in a partially dependent manner, where both statistical properties provide information on location within the plume. "

In conclusion, we believe the reviewers suggested integration mechanism indeed in part underlies the discrimination of intermittency but cannot explain it in full. Both discrimination of integrated concentration and concentration-independent temporal properties are most likely in play.

2) The authors tried to provide an argument for an intermittency by presenting a predicted performance based on concentration Figure S2B. But I could not find explanations of how this is estimated.

Please see our reply to issue 1 where we explain this in detail in broader context of the gainrelated design, test and outcome and interpretation. Additionally, we agree we did not provide a clear explanation in the text, thank you for noticing. We hence added the following:

We added LL232-239: " If mice are solely relying on intensity-integration, then halving the odor concentration (gain=0.5) would be fully equivalent to halving the intermittency at gain=1, as the amount of total absorbed odorant during a trial would be identical in both cases. Figure 2B demonstrates that psychometric curves for gain=0.5 are not right-shifted versions of gain=1 by the expected equivalent doubling of intermittency. Indeed, the intermittency discrimination thresholds shifted much less (0.4, 0.4 and 0.5) than the expected doubling of 0.3 (i.e., 0.6). Further, although there was an effect of gain on behavioral performance, animal performance at 0.5 gain was significantly better than a psychometric curve prediction of animal performance solely based on odor concentration integration accordingly "

3) In general, an animal is computing some combination of variables, and it is extremely difficult to point out which one is most relevant. So far, the focus on intermittency is not motivated.

We set out to test whether mice can discriminate odor intermittency, which is a physical property of plumes. We have shown we accurately presented mice with different intermittencies in stimuli of different gains, and shapes (natural, binary, periodic). We demonstrate they can indeed discriminate all these stimuli's differences in intermittency. We further behaviorally define how mice may discriminate intermittency (partially on temporal suprathreshold means, partially on concentration integration). Last we describe how the olfactory bulb represents intermittency and how this depends on sniffing. We hence maintain the papers focus on intermittency.

We again point to our Discussion where this is addressed head on (L661-681):

" We found that mouse performance on the intermittency discrimination task is not affected by the odor used or frequency of odor whiffs, but is affected by the concentration gain. This shows that intermittency is a temporal property of odor plumes that can be detected independently from other temporal properties, such as whiff frequency. This distinction may be important if different temporal properties provide at least partially independent information about location within the odor plume, as suggested by Jayaram et al. (2022). Other temporal properties that may indicate distance from and composition of an odor source are odor whiff frequency and the temporal correlation of fluctuating odors, both of which mice are capable of detecting (Hopfield, 1991; Schmuker et al., 2016; Ackels et al., 2021; Dasgupta et al., 2022). It is possible that these temporal properties are either used independently or in concert during odor navigation. Additionally, we show that concentration gain has an effect on intermittency discrimination, suggesting that mice are in part using odor concentration integration for intermittency discrimination. While further work needs to be done to explore discrimination of odor stimuli based on odor integration, a plethora of work suggests that rodents can discriminate odor duration and intensity both at the neural and behavioral levels (Rubin et al., 1999; Rospars et al., 2000; Spors et al., 2002; Li et al., 2014; Wojcik et al., 2014; Sirotin et al., 2015; Li et al., 2020). In some odor plumes, odor concentration and odor intermittency both increase as distance from the odor source decreases, indicating that the integral of measured odor would also increase (Connor et al., 2018). It is possible that odor intermittency and odor integration might inform odor source localization in a partially dependent manner, where both statistical properties provide information on location within the plume. "

We hope the reviewer agrees at this stage, if at least sufficiently so.

4) In the previous review I asked to provide the logic for choosing a specific value for intermittency threshold. I did not fully follow their explanation. The authors presented the graph of intermittency as a function of distance (Figure S1A). If the logic that an animal is using an intermittency as a measure of a distance to the source, then the measurement should be done at the interval of intermittencies where it is mostly informative about the distance. Based on the figure, this should be around 0.5, where the slope of intermittency as a function of distance is the steepest. It would be much more reasonable to show an animal's ability to differentiate intermittency around this level. The ability to discriminate intermittency at the level of 0.15 tells us only that an animal can only extract information that the source is far away. This can be potentially done using other cues.

Indeed, based on Figure S1A our discrimination task, by design, is in essence asking if the mice can discriminate a far odor source (low intermittency) from an intermediate-near source (midhigh intermittency) and answers in the affirmative. This approach is reasonable as it addresses the main question of whether mice can discriminate intermittency at all. Additionally, it provides psychometric curves demonstrating discrimination even at relatively close intermittencies (i.e., between far and intermediate distances). Note that this is the first study every to do so in mammals. We agree that many additional and more refined questions follow from our work and we applaud the reviewer for suggesting such. We believe a design exploring what the just noticeable intermittency differences are across the entire proximal distal dimension would be the most informative study at this time, in particular when combined with bulbar and cortical recordings.

Please see our previous responses to your issue 3 as well, repeated below, but I hope our rationale is clearer now.

"Intermittency drops as function of distance from the source (downwind). It also has a close to normal (with kurtosis) distribution across wind, peaking at the center (see e.g. Crimaldi 2002, Connor 2018). So, animals may encounter any and all intermittencies (0-1). Given our Go/NoGo paradigm we had to set a CS-/CS+ boundary. Typically, to generate an adequate psychometric curve using this paradign, either the CS- or CS+ stimuli need to represent a wide range of values of which the animals are required to compare against a narrow range (or single value). Again, bounded by effective behavioral paradigm design, the number of CS+ and CS- trials need to be even in order to appropriately motivate animals to engage in the task. Thus, considering the entire range of intermittency values animals can encounter while navigating through a plume in conjunction with effective behavioral design, we arrived at our chosen values for low and high intermittency.

As you can see in the above figure (and also reviewer #1, comment 2), I=0.15 is roughly at the knee where the monotonic decrease begins to asymptote. This is roughly true for all 3 concentration thresholds. Consequently, I=0.2-0.8 effectively samples the region where intermittency clearly relates to distance to the source, which is where we hypothesize animals."